# Plasma Membrane Ca^2+^ ATPase Isoform 4 (PMCA4) Has an Important Role in Numerous Hallmarks of Pancreatic Cancer

**DOI:** 10.3390/cancers12010218

**Published:** 2020-01-16

**Authors:** Pishyaporn Sritangos, Eduardo Pena Alarcon, Andrew D. James, Ahlam Sultan, Daniel A. Richardson, Jason I. E. Bruce

**Affiliations:** 1Division of Cancer Sciences, School of Medical Sciences, Faculty of Biology, Medicine and Health, University of Manchester, Manchester M13 9PT, UK; pishyaporn.sritangos@manchester.ac.uk (P.S.); eduardo.penaalarcon@postgrad.manchester.ac.uk (E.P.A.); daniel.richardson-3@postgrad.manchester.ac.uk (D.A.R.); 2Department of Biology, University of York, Heslington, York YO10 5DD, UK; andrew.james@york.ac.uk; 3Department of Pharmaceutical Science, College of Pharmacy, Princess Nourah Bint Abdulrahman University, Riyadh, Saudi Arabia; AHMsultan@pnu.edu.sa

**Keywords:** pancreatic ductal adenocarcinoma, plasma membrane Ca^2+^ ATPases 4 (PMCA4), cell migration, apoptosis, cancer metabolism

## Abstract

Pancreatic ductal adenocarcinoma (PDAC) is largely resistant to standard treatments leading to poor patient survival. The expression of plasma membrane calcium ATPase-4 (PMCA4) is reported to modulate key cancer hallmarks including cell migration, growth, and apoptotic resistance. Data-mining revealed that PMCA4 was over-expressed in pancreatic ductal adenocarcinoma (PDAC) tumors which correlated with poor patient survival. Western blot and RT-qPCR revealed that MIA PaCa-2 cells almost exclusively express PMCA4 making these a suitable cellular model of PDAC with poor patient survival. Knockdown of PMCA4 in MIA PaCa-2 cells (using siRNA) reduced cytosolic Ca^2+^ ([Ca^2+^]_i_) clearance, cell migration, and sensitized cells to apoptosis, without affecting cell growth. Knocking down PMCA4 had minimal effects on numerous metabolic parameters (as assessed using the Seahorse XF analyzer). In summary, this study provides the first evidence that PMCA4 is over-expressed in PDAC and plays a role in cell migration and apoptotic resistance in MIA PaCa-2 cells. This suggests that PMCA4 may offer an attractive novel therapeutic target in PDAC.

## 1. Introduction

Pancreatic cancer, particularly pancreatic ductal adenocarcinoma (PDAC), is a commonly diagnosed cancer with the lowest 5-year survival rate of all cancers [1]. The early stages of pancreatic cancer are often asymptomatic resulting in advanced stage diagnosis, with high metastatic tumor burden [2]. Moreover, conventional chemotherapeutics, involving combinations of gemcitabine/nab-paclitaxel, have minimally improved the survival of pancreatic cancer patients over the past three decades [3]. 

Dysregulation of calcium (Ca^2+^) signaling has been reported to facilitate malignancies in multiple types of cancer, including PDAC [4]. Spatiotemporal shaping of Ca^2+^ signaling is critical for regulating numerous physiological processes in all cells [5]. However, this can only be achieved if cytosolic Ca^2+^ is maintained very low within cells. Pathological elevation of cytosolic Ca^2+^, particularly an irreversible increase in [Ca^2+^]_i_ (Ca^2+^ overload), has been associated with apoptotic and necrotic cell death [6,7]. Over-expression of PMCAs, in particular, have been associated with resistance to such Ca^2+^ overload-mediated cell death in PDAC [8,9,10].

Plasma membrane Ca^2+^ ATPases (PMCAs) are major ATP-consuming pumps responsible for Ca^2+^ extrusion from the cells. There are four isoforms of PMCAs (PMCA1–4) encoded by four distinct genes (ATP2B1–4). Complex alternative splicing of these primary ATP2B1–4 gene transcripts generates a diverse array of more than 30 splice variants [11]. PMCA1 is a ubiquitously expressed housekeeping Ca^2+^ efflux pump. Conversely, PMCA2–3 isoforms are ‘rapid response pumps’ which are selectively expressed in excitatory cells such as neurons. PMCA4, albeit ubiquitously expressed [12], has been shown to exhibit a more specific role as a signaling hub for proliferative signaling [13], NFkB nuclear translocation [14], TNF-induced cell death [15] and migration [16]. Many of these signaling pathways are important for the maintenance of cancer hallmarks [17]. Therefore, we hypothesized that PMCA4 may have an important role in numerous cancer hallmarks in PDAC.

Characteristics of PDAC often include high metastatic burden and a metabolic shift towards glycolysis [2,18]. Our previous studies in PDAC cells demonstrated that the PMCA is reliant on glycolytic ATP supply; specific glycolytic inhibitors cut off this ATP supply leading to inhibition of PMCA activity, cytotoxic Ca^2+^ overload and cell death [9,10]. Moreover, in red blood cells and smooth muscle cells, the PMCA has been suggested to have its own privileged glycolytic ATP supply, in which glycolytic enzymes are associated with the plasma membrane in close proximity to the PMCA [19,20,21]. Conversely, key glycolytic enzymes such as phosphofructokinase [22] and pyruvate kinase are inhibited by high ATP concentrations [23]. Since PMCA is a major ATP consumer, such close functional coupling may help to maintain glycolytic flux.

As PMCA4 has been implicated in numerous cancer hallmarks, previous studies had assessed the role of PMCA4 in MDA-MB-231 breast cancer cells [14] and HT-29 colon cancer cells [24]. Although ≥5-fold over-expression of PMCA4 mRNA has been detected in PDAC tumors compared to normal pancreatic ductal epithelial cells [25], there are limited studies that have assessed the specific role of PMCA4 in PDAC.

The calcium signaling machinery represents a rich tapestry of potential therapeutic targets for the treatment of cancers. This is because the spatiotemporal shaping of calcium signals and the remodeling of Ca^2+^ signaling machinery is reported to lead to numerous cancer hallmark responses, including cell proliferation/cell cycle control, apoptosis, migration/invasion, angiogenesis [26] as well as fibrosis and drug resistance that can contribute to poor patient prognosis [27,28]. However, due to the ubiquitous nature of most Ca^2+^ transport pathways, targeting them with specific drugs will most likely lead to adverse effects, unless they are either uniquely over-expressed or exhibit a unique function. The current study identified PMCA4 as a potential candidate as it appears to be almost uniquely over-expressed in MIA PaCa-2 cells similar to human PDAC tumors, making these an ideal model cell line to study the role of PMCA4 in PDAC. Moreover, PMCA4 has an important role in cancer hallmark responses, including migration and apoptosis resistance. This work highlights the importance of PMCA4 in multiple cancer hallmarks and identifies PMCA4 as a potential novel therapeutic target for the treatment of PDAC.

## 2. Results

### 2.1. Expression of the PMCA4 Gene, ATP2B4, Is Correlated with PDAC and Poor Patient Survival

To determine whether alterations in ATP2B1–4 gene expression (encoding PMCA1–4 protein) in PDAC tumor are correlated with poor patient survival, data-mining from two independent open-source databases, Oncomine [29,30] and The Human Protein Atlas [31,32], was performed.

Gene chip microarray data showed that ATP2B4 was predominantly over-expressed (2.65-fold, n = 39, *p* < 4.06^−10^) to a much greater extent than ATP2B1 (1.24-fold, n = 39, *p* < 0.035) in human PDAC tumors versus resected healthy tissue from the tumor margin (Badea et al., 2008). In contrast, expression of both ATP2B2 (−1.44-fold, n = 39, *p* < 1.92^−9^) and ATP2B3 (−1.56-fold, n = 39, *p* < 1.95^−8^) were significantly reduced in PDAC (Figure 1A–E).

Patient survival data was sourced from the cancer genomic atlas–pancreatic adenocarcinoma cohort (TCGA-PAAD). The cohort of PDAC patients was divided into quartiles based on the median-centered ATP2B1–4 tumor expression. Only patients with high expression (>75th percentile) of ATP2B4 had lower survival (hazard ratio = 1.83, n = 45, *p* < 0.04) whereas the expression of ATP2B1 had no effect (Figure 1F,G). Expression of ATP2B2 and ATP2B3 were negligibly detected and could not be correlated with patient survival.

Collectively, these data suggest that elevated ATP2B4 and low ATP2B2–3 expression are representative characteristics of resected PDAC tumors which correlate with poor PDAC patient survival. The implication of this is that PMCA4 may facilitate cancer hallmark responses and thus drive tumorigenicity. However, it must be acknowledged that the lack of any clinical status (i.e., tumor grade and histological status) associated with these datasets makes the interpretation of these results limited and are thus hypothesis generating.

### 2.2. PMCA4 Is the Major PMCA Isoform Expressed in MIA PaCa-2 Pancreatic Cancer Cell Line

Given that high expression of ATP2B4 correlates with poor PDAC patient survival, we sought to determine the expression PMCA1–4 isoforms in PDAC cellular models in order to identify a suitable in vitro PDAC model which reflects this high ATP2B4-expressing characteristic. PDAC cell lines (MIA PaCa-2 and PANC-1) and related non-malignant pancreatic cells (human pancreatic ductal epithelial cells and human pancreatic stellate cells; human pancreatic ductal epithelial (HPDE) and human pancreatic stellate cells (hPSC), respectively), at both protein and mRNA level. MIA PaCa-2 and PANC-1 are cell lines established from the resected pancreatic carcinoma and exhibited epithelial morphology [33,34]. HPDE is a non-transformed human pancreatic ductal epithelial cell line established from HPV E6/E7*-immortalization [35,36]. On the other hand, although not considered to be malignant, hPSC is a primary culture derived from a PDAC tumor resected from a patient who had undergone a Whipple procedure and therefore these cells exhibit classical hallmarks of activated stellate cells [37,38] associated with PDAC progression [39,40].

Using Western blot, relative PMCA4 expression was the highest in both MIA PaCa-2 and hPSC compared to HPDE and PANC-1 (Figure 2A,B). PMCA2 was also detected in MIA PaCa-2 and hPSC to a greater extent than HPDE and PANC-1. However, despite high expression in the mouse brain lysate positive control, PMCA1 and PMCA3 were barely detected in any pancreatic cell lines screened. RT-qPCR further confirmed that ATP2B4 was almost exclusively expressed in MIA PaCa-2 cells with close to undetectable levels of ATP2B1–3 (Figure 2C). In relative terms, PANC-1 cells expressed low levels of ATP2B1–4 isoforms, also consistent with Western blot data. The non-malignant HPDE cells abundantly expressed ATP2B1 and ATP2B4 mRNA which was not necessarily consistent with the Western blot data, suggesting that either the primary antibody was relatively poor at detecting PMCA1 or that PMCA1 mRNA was silenced and not translated into protein in HPDE. hPSC expressed a similar pattern of mRNA expression to that observed in HPDE, albeit relatively lower and more variable than MIA PaCa-2.

It should be noted that the relative ratio of ATP2B4:ATP2B1 in hPSC, HPDE and PANC-1 cell lines were relatively similar (0.6–1.4 fold) whereas MIA PaCa-2 substantially expresses ATP2B4 by 48-fold more than ATP2B1 (*p* < 0.05, Figure 2C). Since MIA PaCa-2 cells almost exclusively expressed PMCA4 at both the protein and mRNA level, this suggests that MIA PaCa-2 represents a good cellular model of PMCA4-overexpressing PDAC that can be taken forward for further functional studies. Moreover, as our previous studies by James, A.D. et al. [9,10] suggested that MIA PaCa-2 cells are more reliant on glycolytically–derived ATP than the related PANC-1 cells, this makes MIA PaCa-2 an ideal cellular model to examine the role of PMCA4 on PDAC cancer hallmarks, particularly the metabolic shift towards glycolysis.

### 2.3. siRNA Knockdown of ATP2B4 mRNA and PMCA4 Protein Expression in MIA PaCa-2 Cells

To test the functional effects of PMCA4 in MIA PaCa-2 cells, a pool of 4 siRNA was used to transiently knockdown the expression of ATP2B4 (siPMCA4). A similar pool of non-targeting siRNA (siNT) was used as a control. PMCA4 knockdown was then confirmed at the protein and mRNA level.

Using Western immunoblotting, results show that PMCA4 protein expression decreased in a time-dependent manner in comparison to the siNT control. PMCA4 expression decreased by at least 70% between 48–96 h post-siPMCA4 treatment (Figure 3A,B, n = 3, *p* < 0.05). At mRNA level, the total ATP2B4 and splice variant ATP2B4b expressions were knocked down by at least 70% after 48 h (Figure 3C,D, n = 4, *p* < 0.05). Moreover, there were no compensatory changes in ATP2B1–3 expression (Appendix A). According to the study by Curry et al., conditions yielding ≥70% PMCA mRNA expression knockdown were sufficient to functionally modulate Ca^2+^ signaling in MDA-MB-231 breast cancer cells [14]. Therefore, siRNA incubations between 48–96 h, yielding ≥70% PMCA4 mRNA expression knockdown, were used for further experiments.

### 2.4. PMCA4 Is the Major Functional Ca^2+^ Efflux Pathway in MIA PaCa-2 Cells

Our previous study demonstrated that PMCAs represent the main mechanism of intracellular Ca^2+^ ([Ca^2+^]_i_) efflux in MIA PaCa-2 cells [9,10]. As PMCA4 is the major PMCA isoform expressed in MIA- PaCa-2, we predicted that PMCA4 is functionally critical for Ca^2+^ efflux. After establishing conditions which yielded ≥70% of PMCA4 expression knockdown, we then wanted to confirm that this led to decreased PMCA activity using our in situ Ca^2+^ clearance assay [9,10].

Cells loaded with Ca^2+^ sensing fura-2 dye were perfused with Ca^2+^-free HEPES-buffered physiological saline solution (HPSS) containing 1mM EGTA and 30 μM cyclopiazonic acid (CPA). CPA is an inhibitor of a sarco/endoplasmic reticulum Ca^2+^-ATPase (SERCA) that depletes endoplasmic reticulum Ca^2+^ storage and activates store-operated Ca^2+^ entry (SOCE) channels. The addition of 20 mM Ca^2+^ HPSS leads to SOCE and a large increase in [Ca^2+^]_i_ which reaches a high steady state. Subsequent removal of external Ca^2+^ allows [Ca^2+^]_i_ clearance, almost exclusively due to PMCA [10] which is assessed and quantified by fitting the falling phase to a single exponential decay to yield a time constant (τ) as a measurement of rate.

We found that knocking down PMCA4 led to profound inhibition of [Ca^2+^]_i_ clearance compared to siNT control (Figure 4A,B). In siPMCA4-treated cells, the mean τ was significantly higher (τ = 472 ± 84.0 s, *p* = 0.0091) when compared to siNT controls (τ = 126.8 ± 22.4 s), suggesting Ca^2+^ clearance rate was significantly reduced as expected (Figure 4C,D). In addition, the transient increase in [Ca^2+^]_i_ following addition of CPA, in zero external Ca^2+^, reflects ER Ca^2+^ leak that is cleared from the cytosol by the PMCA. This was also significantly greater in PMCA4 siRNA-treated cells (area under the curve (AUC) = 525.8 ± 65.9, *p* < 0.001) compared to siNT controls (AUC = 211.6 ± 24.8; Figure 4E,F). These results suggest that PMCA4 expression is required for Ca^2+^ efflux and the loss of PMCA4 significantly impairs Ca^2+^ clearance.

### 2.5. PMCA4 Knockdown Inhibits Cell Migration Independent of Cell Proliferation

PMCA4 associated Ca^2+^ efflux has been reported to be important for cell migration [16] as well as cell growth [41]– both important cancer hallmarks [17]. A gap closure assay was used to assess cell migration. Since gap closure can be influenced by cell proliferation as well as migration, the anti-proliferative reagent mitomycin C (Mit C) was used to prevent cell proliferation from having a confounding effect on gap closure [42,43]. Sulforhodamine B (SRB) assay was used to confirm that mitomycin C successfully inhibited cell growth.

Without Mit C treatment, gap closure attributed to both cell migration and proliferation was relatively similar between siPMCA4 and siNT control (Figure 5A,B). Furthermore, no differences between the growth rate of siPMCA4 and siNT control were observed from SRB assays (Figure 5C). Upon treatment with Mit C, to selectively monitor cell migration, siPMCA4-treated cells exhibited reduced gap closure (65.6 ± 4.4 % closure, *p* < 0.05) compared to siNT controls after 48 h (80.6 ± 3.2% closure; Figure 5A,B). Mit C treatment inhibited cell growth of both siNT and siPMCA4 at 48 h without any noticeable effect on cell viability. However, at 72 h SRB absorbance had declined, suggesting that Mit C was affecting cell viability beyond 48 h. Nevertheless, this suggests that any changes in gap closure in Mit C-treated cells over 48 h represent an effect on cell migration, independent of cell viability or cell proliferation. Overall, our results showed that PMCA4 knockdown significantly inhibited cell migration but had no effect on the growth rate of MIA PaCa-2 cells.

### 2.6. PMCA4 Knockdown Sensitizes MIA PaCa-2 Cells to Apoptosis

Over-expression of PMCAs is thought to confer apoptotic resistance in multiple cancers [14,44]. Therefore, to investigate whether PMCA4 plays a role in apoptotic resistance, siNT and siPMCA4 treated cells were exposed to various stressor reagents and caspase 3/7 cleavage assay was used as a measure of apoptosis. A classical apoptotic inducer, staurosporine (STS), was used as a positive control. Based on our previous studies, PMCA activity is modulated by glycolytic inhibitors but not mitochondrial inhibitors in MIA PaCa-2 cells [10]. Therefore, the stressors used in caspase 3/7 cleavage assay included glycolytic inhibitor (iodoacetate, IAA), mitochondrial inhibitor (oligomycin, OM) and SERCA inhibitor (CPA) known to induce ER stress and Ca^2+^ overload. 

Over a period of 12 h, we found that STS significantly induced caspase 3/7 cleavage in both siNT (23.9 ± 2.1%) and siPMCA4 (37.5 ± 2.7%) compared to DMSO vehicle control (11.5 ± 1.38%; Figure 6A; Appendix A). Our results showed that PMCA4 knockdown cells were significantly more susceptible to STS, CPA, and OM induced apoptosis compared to siNT control. PMCA4 knockdown cells were noticeably sensitized to CPA-induced cell death (52.5 ± 9.5% at 12 h) and significant caspase 3/7 cleavage was observed from as little as 7 h (32.4 ± 7.6% at 7 h; Figure 6A,B). Although CPA is suggested to induce ER stress associated cell death [45], CPA-induced significant apoptosis in siPMCA4 but had no effect on siNT control. Conversely, IAA caused similar levels of apoptosis in both siPMCA4 (40.5 ± 16.0%) and control (44.1 ± 16.9%; Figure 6A). Overall, in comparison to vehicle control, CPA and OM had no effect on siNT control but significantly increased apoptosis in PMCA4 knockdown cells, suggesting that PMCA4 expression knockdown sensitized MIA-PaCa2 PDAC cells to Ca^2+^ and mitochondrial stress.

Although CPA would induce similar ER Ca^2+^ depletion and ER stress in both siNT and siPMCA4 treated cells, our results suggest that cytosolic Ca^2+^ overload was more substantial and irreversible in Ca^2+^ efflux impaired PMCA4 knockdown cells (Figure 6C), facilitating Ca^2+^ overload-associated apoptosis. Using the Ca^2+^ overload assay, we showed that CPA treatment triggered a substantial increase in [Ca^2+^]_i_ which caused fura-2 signal saturation (Rmax), making calibration of [Ca^2+^]_i_ difficult (Figure 6C). Therefore, the magnitude of these [Ca^2+^]_i_ responses was quantified by measuring the area under the curve (AUC) of uncalibrated fura-2 signal ratios (Figure 6D). The AUC of siPMCA4-treated cells was significantly higher than the siNT control by 1.5-fold. Although Ca^2+^ calibration could not be applied to CPA responses, it was possible to calibrate and compare the resting [Ca^2+^]_i_. Interestingly, the resting [Ca^2+^]_i_ of siPMCA4 cells (286 ± 68 nM) was 3.6-fold higher than siNT controls resting [Ca^2+^]_i_ (79 ± 12 nM). (Figure 6E) These results suggest that PMCA4 knockdown cells had inherently higher resting [Ca^2+^]_i_ due to impaired [Ca^2+^]_i_ efflux.

### 2.7. PMCA4 Knockdown Alters Metabolic Flexibility during Ca^2+^ Overload

Ca^2+^ is a known modulator of both glycolytic [46] and mitochondrial metabolism [47]. Moreover, we found that knocking down PMCA4 unexpectedly sensitized cells to mitochondrial ATP synthase inhibitor (OM), suggesting that mitochondrial respiration was altered in the absence of PMCA4. As PMCA is a crucial Ca^2+^ efflux mechanism that is also functionally reliant on glycolytic ATP in PDAC cells [9,10], we hypothesized that PMCA4 knockdown may alter the metabolic phenotype of MIA PaCa2 PDAC cells.

The Seahorse XFe96 Analyzer was used to monitor real-time changes in oxygen consumption rate (OCR) and extracellular acidification rate (ECAR), indicative of mitochondrial respiration (oxidative phosphorylation; OXPHOS) and glycolysis, respectively. The current study compared changes in both OXPHOS and glycolytic metabolism of siNT control versus siPMCA4 cells unstimulated (unstressed) and Ca^2+^ overload conditions (Ca^2+^ stressed). CPA was added prior to mito/glycolysis stress test to induce Ca^2+^ overload stress, subsequently forcing PMCA4 to function at its maximum capacity. 

To investigate the differences between the mitochondrial respiration of siNT control versus siPMCA4, key parameters of mitochondrial functions were measured using the Seahorse mito stress test (Figure 7A,D). In the absence of Ca^2+^ stress, PMCA4 knockdown had no effect on any mitochondrial respiration parameters. However, we observed a 1.8-fold increase in mitochondrial membrane potential (ΔΨm), indicative of a larger driving force for mitochondrial ATP synthesis (Appendix A).

However, CPA-induced Ca^2+^ stress appeared to reduce maximum respiration capacity in siNT-treated control cells, although this did not reach statistical significance. Nevertheless, this Ca^2+^ stress-induced reduction in maximal respiration capacity was abolished in siPMCA4 treated cells (Figure 7D). We next tested the effect of Ca^2+^ stress on glycolysis using the glycolysis stress test to measure changes in ECAR which reflect glycolytic flux (Figure 7E,H). Under basal conditions, PMCA4 knockdown had no effect on all glycolytic parameters. Under Ca^2+^ stress, the glycolytic reserve was significantly lower in siPMCA4 compared to siNT, although the change was small and may not be physiologically relevant (Figure 7H). It should be noted that MIA PaCa-2 cells had a low glycolytic reserve which is indicative of metabolic reliance on glycolysis.

Finally, simultaneous analyses of mitochondrial and glycolytic-linked ATP production rates were derived from ECAR and OCR based on the Agilent Seahorse XF real-time ATP rate assay calculations [48]. The results revealed that over 70% of MIA PaCa-2 ATP production rate was glycolytically linked and this pattern did not change under Ca^2+^ stress condition (Appendix A). This suggests that MIA PaCa-2 cells are heavily dependent on aerobic glycolysis regardless of PMCA4 expression. Under Ca^2+^ stress, even though the ATP production rate did not change in siNT control, PMCA4 knockdown cells showed significantly increased glycolysis-linked ATP production rate (Figure 7I).

Overall, we found that PMCA4 expression did not alter the basal metabolic phenotype of MIA- PaCa-2 cells. Under Ca^2+^ stress, siNT control showed decreased mitochondrial respiration capacity and increased the shift in glycolytic reserve while demonstrating unaltered ATP production rate. In contrast, all CPA effects observed were abolished when PMCA4 was knocked down and enhanced glycolytic-linked ATP production rate was observed. Nevertheless, these changes were relatively minor considering the levels of Ca^2+^ stress and metabolic stress and unlikely to be of major phenotypical relevance.

## 3. Discussion

The current study has provided insights into the functional importance of PMCA4 on multiple cancer hallmarks in PDAC. Through data-mining of available databases, we found that over-expression of ATP2B4 and reduced expression of ATP2B1–3 are likely characteristics of PDAC tumors [30] that are correlated with poor PDAC patient survival prognosis [31,32]. ATP2B4 mRNA, in particular, has been reported to be over-expressed by 5-fold in PDAC tumors compared to the healthy tissue margin [25]. Therefore, we consider ATP2B4 to be an important PMCA encoding gene linked with poor PDAC patient survival. We then identified MIA PaCa-2 cell line as an ideal cellular model for PMCA4-overexpressing PDAC based on the almost exclusive expression of PMCA4. 

The specific spatiotemporal patterns of Ca^2+^ signals regulate diverse cellular processes. However, this can only be achieved if resting [Ca^2+^]_i_ is maintained low (~100 nM) by Ca^2+^ efflux machinery, including the PMCA [49]. Our previous study demonstrated that PMCA is the main Ca^2+^ efflux mechanism in PANC-1 and MIA PaCa-2 PDAC cells [10]. Among the four PMCA isoforms, PMCA1 and 4 are regarded as ubiquitously expressed PMCA isoforms. Although total PMCA1 knockout is embryonic lethal, PMCA4 knockout mice are generally normal except for male infertility [50,51]. This is because there is considerable functional redundancy in most cell types where PMCA1 is expressed, and is thus able to compensate for the loss of PMCA4 [52]. However, as sperm predominantly expresses PMCA4 [53], the knockdown of PMCA4 resulted in inhibited sperm motility and thus male infertility. 

Extrapolation to PDAC cells which predominantly expresses PMCA4, it could be argued that PMCA4 may represent a novel therapeutic target and thus selective PMCA4 inhibition may also impair cell function and sensitize high PMCA4-expressing PDAC cells to apoptosis. The present study reveals that PMCA4 is predominantly expressed and has an almost exclusive role in Ca^2+^ efflux in MIA PaCa-2 cells. Consistent with previous observations in mice sperm [50,51] and Jurkat cells [54], we showed that knocking down PMCA4 expression significantly impaired Ca^2+^ clearance, subsequently elevating resting [Ca^2+^]_i_. Moreover, PMCA4 mediated Ca^2+^ signaling, in particular, has also been implicated in multiple cancer hallmarks including [16], cell cycle progression [41] and cell death [14]. However, it must be noted that PMCA4 is not over-expressed in all PDAC cells (e.g., PANC-1) and PMCA4 abundance may depend on cell differentiation status and confluency [24]. Moreover, PDAC cells are usually well or moderately differentiated histologically, which may influence PMCA4 abundance. Nevertheless, Kaplan–Meier survival analysis suggests that PMCA4 over-expression correlates to poor patient survival. Therefore, MIA PaCa-2 cells constitute a well-suited model for an eventual “PMCA4 high expression” molecular subtype of PDAC. Furthermore, by extrapolation MIA PaCa-2 cells may also represent a good cellular model for testing/screening novel putative PMCA4-specific inhibitors that would be predicted to improve PDAC patient survival.

More than 40% of PDAC patients are diagnosed with metastasis [55,56] which in turn is dependent on cell migration and invasion. Therefore, it could be argued that migration is an extremely important cancer hallmark for PDAC. Cell migration is a process regulated by discrete spatiotemporal Ca^2+^ signals which modulate the assembly and disassembly of the cytoskeleton and focal adhesion complexes at the leading and trailing edges of migrating cells [57,58]. In human umbilical vein endothelial cells, enrichment of PMCA4 at the migrating front is suggested to maintain lower Ca^2+^ gradient (~30 nM) [16], facilitating focal adhesion assembly [57]. Conversely, impaired PMCA4 activity has been implicated in compromised cell migration and motility [16,50,59]. Consistent with previous findings in rabbit corneal epithelium cells [59], the current study finds that PMCA4 plays an important role in PDAC cell migration and this migration is inhibited in Ca^2+^ clearance-impaired PMCA4 knockdown cells. 

Although cell proliferation is another cancer hallmark known to be modulated by Ca^2+^ signaling and PMCA4 is reported to be important for G1-phase progression and cell proliferation in mice-derived vascular smooth muscle cells [41], we found that knocking down PMCA4 did not affect MIA PaCa-2 proliferation rate nor cell viability. However, as PMCA4 knockdown is not 100% in this study, it is plausible that any residual PMCA4 expression, or expression of any other PMCA isoforms-even though very low, may be sufficient to maintain Ca^2+^ homeostasis and thus cell viability. Similar to our data, PMCA4 knockdown in MDA-MB-231 breast cells had no effect on cell cycle, proliferation and viability [14,60]. 

Apoptosis resistance is a key cancer hallmark responsible for insensitivity to PDAC therapeutic treatment [61]. Elevated [Ca^2+^]_i_ can induce apoptotic cell death by Ca^2+^ overload, the subsequent release of cytochrome C from the mitochondria and activation of pro-apoptotic caspases. Over-expression of PMCAs in cancer prevents [Ca^2+^]_i_ overload under stress stimuli and subsequently leads to apoptotic resistance [62]. In MDA-MB-231 breast cancer cells, PMCA4 and PMCA1 distinctly mediate resistance to apoptotic and necrotic cell death, respectively [14]. Consistent with this observation, the current study shows that PMCA4 knockdown sensitized MIA PaCa-2 cells to apoptotic inducers, particularly CPA-mediated Ca^2+^ overload. This suggests that PMCA4 is required for Ca^2+^ overload associated with apoptotic resistance. 

Both mitochondrial and glycolytic metabolism are known to be modulated by Ca^2+^ [46,63,64]. The activity of key rate limiting glycolytic enzyme phosphofructokinase 1 (PFK) is modulated by Ca^2+^-dependent calmodulin binding [46]. Conversely, Ca^2+^ is reported to promote the activity of the tricarboxylic acid cycle (TCA) dehydrogenases required for mitochondrial respiration [65]. On the other hand, PMCA may play a role in the metabolic shift towards glycolysis. In red blood cells and smooth muscle cells, multiple glycolytic enzymes are localized to the plasma membrane, providing a privileged ATP supply to the PMCA [6,19]. Moreover, it has been suggested that PMCA preferentially uses plasma membrane associated glycolytic ATP [20,21]. Therefore, we hypothesized that PMCA4 may be functionally involved in PDAC metabolism. However, we found that PMCA4 expression had no effect on both basal mitochondrial and glycolytic metabolism. 

Although PMCA4 knockdown did not alter mitochondrial respiration and ATP production rate, we observed a significant increase in resting cytosolic Ca^2+^ and mitochondrial membrane potential (ΔΨm). Similar to our observations, a study in PC12 neuronal cells showed that PMCA2 and PMCA3 knockdown led to increased resting ΔΨm [66]. We suspect that PMCA4 knockdown cells adapted to elevated resting [Ca^2+^]_i_ by maintaining a higher ΔΨm to avoid mitochondrial membrane depolarization which subsequently triggers apoptotic cells death [5].

Under Ca^2+^ stress, the reduced mitochondrial respiration in PMCA4 expressing MIA PaCa-2 cells may be related to SERCA inhibitor-mediated mitochondrial depolarization [67]. However, the mechanism of this impaired mitochondrial function remains unclear. In contrast, PMCA4 knockdown cells showed enhanced glycolysis-linked ATP production rate which suggests escalating demand for ATP. As PMCA4 knockdown cells maintained a higher resting ΔΨm, this increased glycolytic-linked ATP production rate during Ca^2+^ stress may be associated with paradoxical hydrolysis of ATP by the ATP synthase to maintain ΔΨm [68], or apoptotic associated ATP demand [69]. However, it should be noted that potential observations of ATP synthase functioning in reverse may be minimized due to the use of oligomycin (ATP synthase inhibitor) in Seahorse assays.

In the context of the tumor microenvironment in vivo, the over-expression of PMCA4 function combined with the highly glycolytic phenotype in PDAC is likely to have an even greater role in apoptosis resistance and tumor progression. Specifically, the tumor microenvironment is notoriously highly acidic, due to the highly glycolytic phenotype and lactic acid efflux [70]. This acidic microenvironment facilitates matrix metalloproteinases and thus migration and invasion [71], but also inhibits immune surveillance [72]. Specifically, tumor acidity acts as a broad immune escape mechanism by which cancer cells, inhibit anti-tumor immune effectors (including T cells, NK cells and crucial antigen-presenting dendritic cells), while simultaneously promoting the immunosuppressive properties of regulatory T cells and myeloid cells. However, in the context of the current study, the acidic microenvironment may also specifically promote PMCA4 activity and thus further promote cell survival. This is because PMCAs are ATP-driven Ca^2+^/H^+^ exchangers and thus extracellular acidification facilitates Ca^2+^ efflux and alkalinization inhibits Ca^2+^ efflux [73,74]. This means that a highly glycolytic PDAC cell phenotype not only provides a privileged ATP supply to PMCAs but also further accentuates PMCA activity by promoting lactic acid efflux and extracellular acidification. This therefore means that within the context of a hypoxic acidic tumor microenvironment in vivo, over-expression of PMCA4 may have an even more pro-survival, pro-migratory phenotype than observed in our 2D cell culture model. This also suggests that targeting PMCA4 using novel and specific inhibitors may be especially effective in combination with drugs that target tumor acidification. These include glycolytic inhibitors [75], lactic acid/monocarboxylate transporter (MCT) inhibitors [76], lactate dehydrogenase (LDH) inhibitor [77,78] and hypoxia inducible factor (HIF1α) inhibitors [79]. In addition, these drugs may also have a synergistic effect when combined with other novel immunotherapies such as checkpoint inhibitors [80].

Taken together, PMCA4 has a distinct role in Ca^2+^ clearance, cell migration and apoptotic resistance in PDAC. PMCA4 knockdown alone did not alter cell viability in PDAC cells despite the elevated resting [Ca^2+^]_i_. However, loss of PMCA4 expression sensitizes PDAC cells to Ca^2+^ overload-associated apoptotic cell death. As PDAC is notoriously insensitive to current clinical therapies, acquiring the ability to selectively sensitize PMCA4-overexpressing PDAC cells to apoptotic-inducing chemotherapy is crucial. Therefore, targeting PMCA4 may potentially be beneficial as a therapeutic adjuvant which selectively sensitizes PDAC cells to currently available clinical therapy while sparing healthy tissues.

## 4. Materials and Methods

### 4.1. Cell Culture

MIA PaCa-2 and PANC-1 were purchased from ATCC. Human pancreatic ductal epithelial cells (HPDE) and human pancreatic stellate cells (hPSC) were a kind gift from Diane Simeone (University of Michigan) and David Yule (University of Rochester), respectively. MIA PaCa-2, PANC-1, and hPSC were cultured in DMEM media supplemented with 10% fetal bovine serum and 1% penicillin/streptomycin (Sigma–Aldrich, Gillingham, UK). HPDE were cultured in Keratinocyte-SFM media (Fisher Scientific, Loughborough, UK). All cells were cultured under 5% CO_2_ (g), at 37 °C. Mycoplasma contaminations were screened by DAPI/Hoechst 33,342 staining [81] using a Zeiss Axioimager.D2 upright microscope using a 100×/1.4 Plan Apochromat (Oil) on a DAPI bandpass filter set (Carl Zeiss Ltd., Oberkochen, Germany) or submitted to the University of Manchester FBMH Media Order services for PCR-based detection method.

### 4.2. Chemicals and Reagents 

All chemicals and solvents were purchased from Sigma–Aldrich (Gillingham, UK), unless otherwise specified.

### 4.3. Data-Mining 

Badea Pancreas (2008) gene chip microarray data was obtained from www.oncomine.org, January 2019 (ThermoFisher Scientific, Waltham, *MA*, USA). TCGA-PAAD Kaplan–Meier survival data were obtained from www.proteinatlas.org, version 18.1, January 2019. (Appendix A) All raw data obtained from data-mining were re-plotted and analyzed using GraphPad Prism version 8.

### 4.4. SiRNA Knockdown of PMCA4 Expression

PMCA4 expression knockdown was achieved using 0.1% DharmaFect1 transfection reagent and 25 nM ON-TARGETplus pool siRNA targeting human ATP2B4 mRNA (siPMCA4; catalogue # L-006118-00-0010). ON-TARGETplus non-targeting pool siRNA (siNT; catalogue # D-001810-10-20) was used as a control. (Dharmacon, CO, USA) Transfection conditions that yielded ≥70% expression knockdown between 48–96 h were used in the study. mRNA and protein knockdowns were confirmed by RT-qPCR and Western immunoblotting, respectively.

### 4.5. Quantitative RT-qPCR 

Pre-designed KiCqStart^®^SYBR^®^ Green primers targeting ATP2B1–4 (PMCA1–4) and 18S rRNA were purchased from Sigma–Aldrich. Primers targeting ATP2B4 transcript variant 2 (PMCA4b) were purchased from Sigma–Aldrich. All primer sequences used were in Table 1:

TRIzol Plus mRNA purification kit (ThermoFisher Scientific, Waltham, MA, USA) was used for mRNA extraction. mRNA samples were treated with RNAse-free DNAse (Promega, Hampshire, UK) then reverse transcribed using Taqman^TM^ reverse transcription kit (ThermoFisher Scientific, Waltham, MA, USA). qPCR was performed using POWER SYBR green master mix (Fisher Scientific, Loughborough, UK) on StepOnePlus^TM^ Real-time PCR system (ThermoFisher Scientific, Waltham, MA, USA). The relative quantification of designated mRNA was achieved by S18 rRNA normalization (2^−ΔCT^) then; if possible, further comparisons were made between siRNA knockdown conditions and siNT control (2^−ΔΔCT^).

### 4.6. Western Immunoblotting 

Cell lysates were prepared in RIPA buffer supplemented with cOmplete, EDTA-free protease inhibitor cocktail and PhosSTOP^TM^ phosphatase inhibitor (Sigma–Aldrich). Cell lysate protein concentrations were assessed by Bradford assay prior to gel loading. 5–10 μg proteins were resolved using either NuPAGE 4–12% Bis-Tris gel or NuPAGE 3–8% Tris-acetate gel (ThermoFisher Scientific, Waltham, MA, USA) then transferred onto either PDVF membranes (Bio-Rad, Hertfordshire, UK) or nitrocellulose membranes (0.45 μm nitrocellulose; Amersham, Sigma–Aldrich) using TransBlot® Turbo system (Bio-Rad, Hertfordshire, UK). Immunoblotting was performed using Pan-PMCA (1:1000; clone 5F10, catalogue #MA3-914), PMCA1 (1:1000, catalogue #PA1-914), PMCA2 (1:1000, catalogue #PA1-915), PMCA3 (1:1000, catalogue #PA1-916) and PMCA4 (1:1000; clone JA9, catalogue #MA1-914) primary antibodies purchased from ThermoFisher Scientific (Waltham, MA, USA). β-actin (1:5000; catalogue #3700; Cell Signaling Technology, Danvers, MA, USA) was used as a loading control. Secondary anti-rabbit (1:2000; catalogue #7074) and anti-mouse (1:2000-1:5000; catalogue #7076) antibodies were purchased from Cell Signaling Technology. Protein bands were imaged, assessed for overexposure, and quantified using the ChemiDoc (Fisher Scientific, Loughborough, UK). No overexposed protein bands of interest were used for analysis.

### 4.7. Sulforhodamine B (SRB) Cell Viability Assay 

Cells were seeded at 5000 cells/well into 5 Corning 96-well plates (Sigma–Aldrich, Gillingham, UK). Cells were treated with 25 nM siNT or siPMCA4 then fixed with 10% trichloroacetic acid at 0, 24, 48, 72, and 96 h post-siRNA treatment. Fixed cells were stained with 0.057% SRB and the dye was resuspended with 10 mM Tris, pH 10.5 according to the protocol by Vichai, V et al. [82]. Protein content was quantitated using Synergy HT microplate reader (BioTEK, Swindon, UK) at 540 nm absorbance. Blank wells, containing no cells, were treated in the same manner as wells containing cells. Blank well values were subtracted from experimental readings to eliminate background noise. 

### 4.8. PMCA Activity Assay (Ca^2+^ Clearance)

Cells seeded onto 16 mm circular glass coverslips (VWR, Leicestershire, UK) were incubated with either siNT or siPMCA4 for 48 h. Cells were loaded with 4 μM fura-2 Ca^2+^ dye (TEFLabs, Austin, TX, USA) for 40 min then further equilibrated in HEPES-buffer physiological saline solution (HPSS) for 20 min. Dye-loaded cells were mounted onto a perfusion chamber attached with gravity-fed perfusion system (Harvard apparatus, Holliston, MA, USA) then perfused with Ca^2+^-free HPSS containing 1 mM EGTA and 30 μM CPA for 20 min. This led to ER Ca^2+^ store depletion and activation of store-operated Ca^2+^ entry (SOCE) channels. Therefore, subsequent perfusion of cells with HPSS containing high external Ca^2+^ (20 mM) led to a marked Ca^2+^ entry and increase in cytosolic Ca^2+^ ([Ca^2+^]_i_) which reached a short-lived steady state due to a balance between Ca^2+^ entry and Ca^2+^ efflux. Therefore, subsequent removal of external Ca^2+^, by perfusion with Ca^2+^-free/EGTA-containing HPSS, allowed [Ca^2+^]_i_ clearance to be observed and assessed. The addition of such high external Ca^2+^ was necessary such that [Ca^2+^]i clearance rate could be assessed over a much greater dynamic range of [Ca^2+^]_i_. These methods have been fully characterized in our previous studies [9,10]. Fluorescent signals were acquired at 340/380 nm excitation at 50 ms exposure and the emitted light (510 nm) was separated using a 400 nm dichroic with 505LP filter. Emitted light was detected by CoolSNAP HQ interline progressive-scan CCD camera (Roper Scientific Photometrics, Tucson, AZ, USA). Background-subtracted images were collected at 5 sec intervals on a Nikon TE2000 inverted microscope fitted with ×40 oil immersion objective, 1.3 numerical aperture. Images were acquired using MetaFluor software (Molecular Devices, San Jose, CA, USA). Relative [Ca^2+^]_i_ clearance was determined using single exponential decay fit and results expressed as a time constant (τ). (Extended methods in Appendix A)

### 4.9. Ca^2+^ Overload Assay

Fura2-loaded cells were perfused with HPSS until the 340/380 nm fluorescence ratio stabilized. HPSS containing 30 μM CPA was perfused for 15 min then rinsed with CPA-free HPSS for another 15 min. 510 nm emitted light was imaged using the same method as described in 4.8. Background-subtracted images were collected at 5 sec intervals (×40 oil immersion objective, 1.3 numerical aperture) through the MetaFluor software (Molecular Devices, San Jose, CA, USA). All [Ca^2+^]_i_ calibrations were calculated using the following equation: [Ca^2+^] nM = 541.6 nM * ((Ratio−0.47)/(2.101−Ratio)). (Appendix A) [Ca^2+^]_i_ was quantified by the area under the curve (AUC) as previously described by James, A.D., et al. [9,10].

### 4.10. Cell Migration 

Sterile 2-well culture inserts (Ibidi GmbH, Gräfelfing, Germany) were placed into 12 well culture plates. Cells were seeded at 50,000 cells into each side of the Ibidi insert then treated with designated siRNA for 24 h. Ibidi inserts were removed to create ‘cell-free gap’. Mitomycin C (1 µg/ml; Tocris, Abingdon, UK) was added to designated treatment conditions to prevent cell proliferation from confounding the cell migration results. Regions of interest (cell-free gap area) were immediately defined after Ibidi insert removal through CellSens software (Olympus, Essex, UK). Defined regions of interest were imaged at 0, 24, and 48 h. Images were collected on an Olympus IX83 inverted microscope using a 4x objective and captured using an Orca ER camera (Hamamatsu, Hertfordshire, UK). Images were then processed, and cell-free area was quantified using Fiji ImageJ (http://imagej.net/Fiji/Downloads; Appendix A). % Gap closure was calculated as follow: % gap closure = ((gap area at 0 h−gap area at × h) ÷ gap area at 0 h) × 100

### 4.11. Caspase3/7 Apoptosis Assay 

Cells pre-treated with siRNA for 48 h were treated with apoptotic inducers (STS, OM, CPA, IAA). 5 μM of caspase 3/7 green apoptosis reagent (IncuCyte, Ann Arbor, MI, USA) and 625 nM Nuclear-ID Red (Enzo Life Sciences, Exeter, UK) were added 30 minutes prior to imaging. Cells were imaged once every hour, from 0–12 hour, on the IncuCyte, using 10× lens. Background-subtracted images were analyzed using IncuCyte Zoom software and data are presented as % apoptosis [(green cell count/image ÷ red cell counts/image) × 100].

### 4.12. Seahorse Live-Cell Metabolic Assays

Cells seeded were incubated for 48 h in either siNT or siPMCA4. Seahorse cartridges were hydrated according to manufacturer instructions. Low phenol-red DMEM, pH 7.4 media was used for all Seahorse-based assays. Either vehicle control (<0.1% DMSO) or 30 µM CPA were pre-injected in addition to the sequentially designated reagents for the mito stress (e.g., 1.5 μM oligomycin, 0.5 μM FCCP, and 0.5 μM rotenone and antimycin) and glycolysis stress test (e.g., 10 mM glucose, 1 μM OM, and 50 mM 2-deoxyglucose). Seahorse data obtained were analyzed using Wave analysis software (Agilent Technologies, Santa Clara, CA, USA). Seahorse XF real-time ATP rate equations were used to convert basal and oligomycin treated OCR and ECAR measurements (derived from mito stress test; Appendix A) into mitochondrial and glycolytic ATP production rates [48].

### 4.13. Statistics 

All data were obtained from at least three independent experiments, each experiment comprising of at least three replicates per treatment condition. Data analysis was performed using Prism 8 software (GraphPad, San Diego, CA, USA). Data are presented as means ± SEM. All data were tested for normality using the Sharpiro–Wilk test. Parametric comparisons were performed using either Student’s t-test or analysis of variance (ANOVA) with Tukey post-test. Non-parametric comparisons were performed using Mann–Whitney U-test or Kruskal–Wallis with Dunn’s multiple comparisons test. Two-factor comparisons were performed using Two-way ANOVA with Dunnett’s post-test. Statistical significance is defined as *p* < 0.05.

## 5. Conclusions

Over-expression of PMCA4 is observed in patient PDAC tumors and is correlated with poor patient survival. The current study showed that MIA PaCa-2 cells almost exclusively express PMCA4 and represent a good model to examine the role of PMCA4 in PDAC. siRNA-mediated knockdown of PMCA4 expression, which resulted in impaired Ca^2+^ clearance, impaired PDAC cell migration and sensitized PDAC cells to Ca^2+^-mediated apoptosis. However, PMCA4 knockdown had minimal effects on cell viability and proliferation. This suggests that PMCA4 expression facilitates cell migration and apoptotic resistance in PDAC cells. Therefore, targeting PMCA4 over-expression in PDAC may selectively sensitize PDAC cells to apoptosis while sparing healthy tissues which express multiple PMCA isoforms.

## Figures and Tables

**Figure 1 cancers-12-00218-f001:**
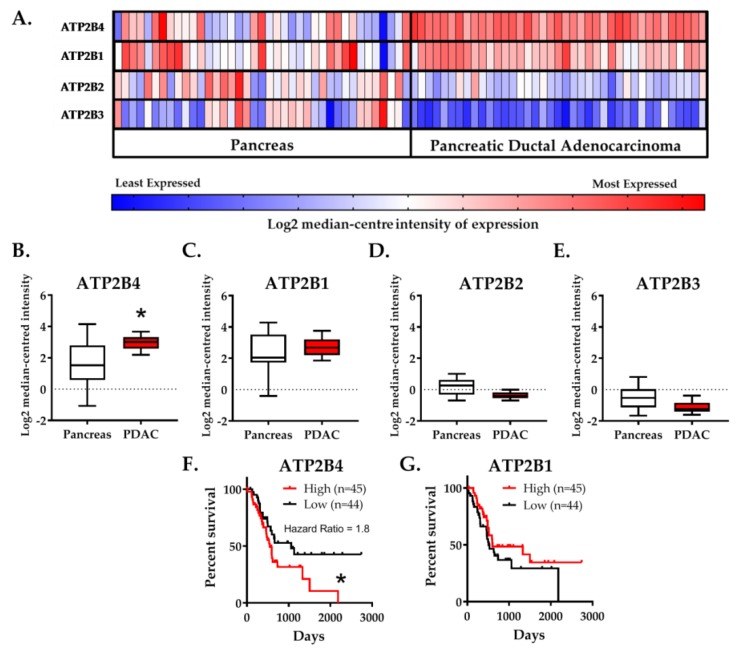
Elevated PMCA4 mRNA expression (ATP2B4) in PDAC is correlated with low patient survival. (**A**–**E**) Badea Pancreas (2008) gene chip microarray data, comparing resected PDAC tumor and healthy pancreatic tissue obtained from matched tumor margin (n = 39), was obtained from Oncomine open-source database. (**A**) Heat map of ATP2B1–4 gene expression in healthy pancreatic tissue and PDAC tumor (n = 39). Heat map colors, ranging from least expressed (blue) to most-expressed (red), depicts relative Log2 median-centered intensity within rows. Heat map colors cannot be compared between rows. Gene expression based on the Log2 median-centered intensity of (**B**) ATP2B4, (**C**) ATP2B1, (**D**) ATP2B2 and (**E**) ATP2B3 are individually presented as box and whisker plots. The whiskers indicate 10–90 percentile of the data range. Statistical comparison between PDAC and healthy pancreas tissue were analyzed using Wilcoxon matched-pairs sign rank test. (**F**,**G**) PDAC patient survival data were sourced from TCGA-PAAD (n = 176), through The Human Protein Atlas database (January 2019, www.proteinatlas.org). The cohort of 176 PDAC patients was divided into quartiles based on the median-centered gene expression (fragments per kilobase of transcript per million mapped reads; FPKM) into either low (25 percentile) and high (75 percentile) gene expression. Kaplan–Meier survival curves correlating the survival of PDAC patients to the low (black) or high (red) expression of (**F**) ATP2B4 and (**G**) ATP2B1. The entire survival outcome curve of the high and low ATP2B4 expressions were used for statistical analysis; the survival outcomes of each group were compared using a log-rank test (Mantel-Cox test). * represents statistical significance where *p* < 0.05.

**Figure 2 cancers-12-00218-f002:**
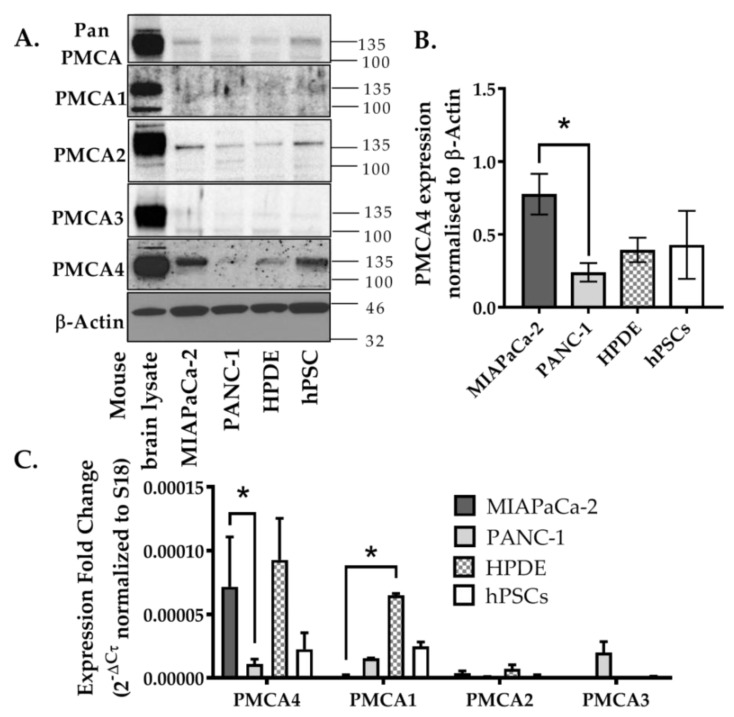
Expression of PMCA isoforms in multiple pancreatic cell lines. (**A**) Representative Western immunoblot showing the relative protein expression of total/pan-PMCA and PMCA isoform 1–4 in pancreatic cancer (MIA PaCa-2 and PANC-1) and non-malignant pancreatic cells (human pancreatic ductal epithelial (HPDE) and human pancreatic stellate cells (hPSC)). Mouse brain lysate was used as a positive control for PMCA expressions and β-Actin was used as a protein loading control. (**B**) PMCA4 protein expression in each cell line was quantified from Western blot bands and normalized to β-Actin housekeeping protein. (**C**) The relative expressions of ATP2B1–4 (PMCA1–4 mRNA) in each cell line were quantified by RT-qPCR. Data are expressed as relative mRNA expression normalized to corresponding S18 rRNA controls (2^−ΔCτ^). Statistical comparisons were made using the Kruskal–Wallis test with Dunn’s multiple comparison test and two-way analysis of variance (ANOVA) with Dunnett’s multiple comparison test. Data are expressed as mean ± SEM. (n = 4–5, 4 replicates per treatment condition). * represents statistical significance where *p* < 0.05.

**Figure 3 cancers-12-00218-f003:**
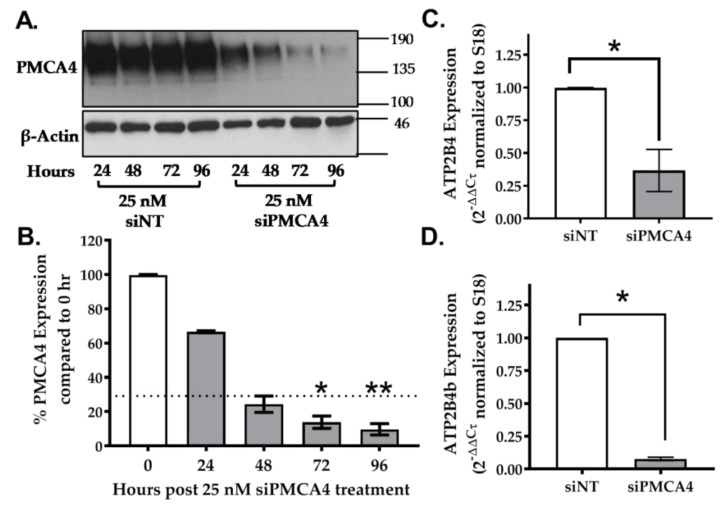
siRNA knockdown of PMCA4 in MIA PaCa-2 PDAC cells. (**A**) Representative Western immunoblotting comparing PMCA4 protein expression in MIA PaCa-2 cells after 24–96 h of treatment with siNT (control) and siPMCA4 (PMCA4 knockdown). (**B**) PMCA4 protein expression of siPMCA4 treated cells was quantified from Western blot bands and normalized to β-Actin (n = 3). Statistical comparison was made using the Kruskal–Wallis test with Dunn’s multiple comparison test, comparing 0-hour control to 24–96 h post-siPMCA4 treatment conditions. (**C**) The relative expression of ATP2B4 (PMCA4 mRNA) and (**D**) ATP2B4b (PMCA4b mRNA) were examined after 48 h of siRNA treatment by RT-qPCR. The expressions of target mRNA were normalized to S18 rRNA and expressed as 2^−(ΔΔCτ)^. Data are shown as mean ± SEM. (n = 4, 4 replicates per treatment condition). Comparisons were made between siNT control and siPMCA4 treated cells at matching time points post-drug treatment using Mann–Whitney U-test (Unpaired, non-parametric rank test). * and ** represents statistically significant difference where *p* < 0.05 and *p* < 0.01, respectively.

**Figure 4 cancers-12-00218-f004:**
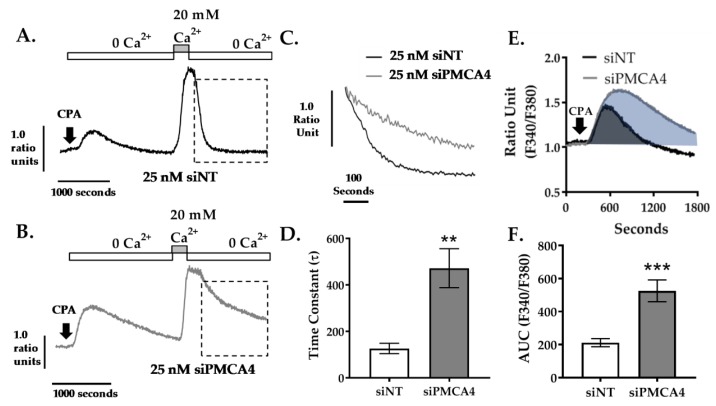
PMCA4 knockdown in MIA PaCa-2 cells reduces PMCA-mediated Ca^2+^ clearance. MIA PaCa-2 cells were treated with non-targeting siRNA (siNT) or siRNA targeting PMCA4 mRNA (siPMCA4) for 48–72 hrs prior to performing an in situ Ca^2+^ clearance assay using fura-2 ratiometric dye. Representative in situ [Ca^2+^]_i_ traces of MIA PaCa-2 incubated in either (**A**) 25 nM siNT or (**B**) 25 nM siPMCA4 are shown. Cells were perfused with Ca^2+^-free + 1mM EGTA HPSS containing 30 μM CPA (white box) to induce ER intracellular Ca^2+^ storage depletion. Cells were then treated with HPSS containing 20 mM Ca^2+^ and 30 μM CPA (grey box) to induce store-operated Ca^2+^ entry. PMCA-mediated Ca^2+^ efflux is observed by subsequent removal of extracellular Ca^2+^ (Ca^2+^-free HPSS). (**C**) Expanded time course showing Ca^2+^ clearance phase (as indicated by dashed box in A.) for siNT vs. siPMCA4 are superimposed. (**D**) The rate of [Ca^2+^]_i_ clearance was fitted to a single exponential decay and the time constant (τ) was determined. (**E**) Expanded time course of the initial CPA-induced increase in [Ca^2+^]_i_ for siNT (black) vs. siPMCA4 (grey) treated cells are shown. Shaded areas indicate the baseline-subtracted area under the curve (AUC). (**F**) Data were quantified by measuring the AUC over 1800 sec. Data are shown as mean ± SEM. (n = 5, at least 50 individual cells were analyzed per treatment condition). Comparisons were made between siNT control and siPMCA4 treated cells at matching time points post-drug treatment using Mann–Whitney U-test (Unpaired, non-parametric cumulative distribution test). ** and *** represents statistically significant difference where *p* < 0.01 and *p* < 0.001, respectively.

**Figure 5 cancers-12-00218-f005:**
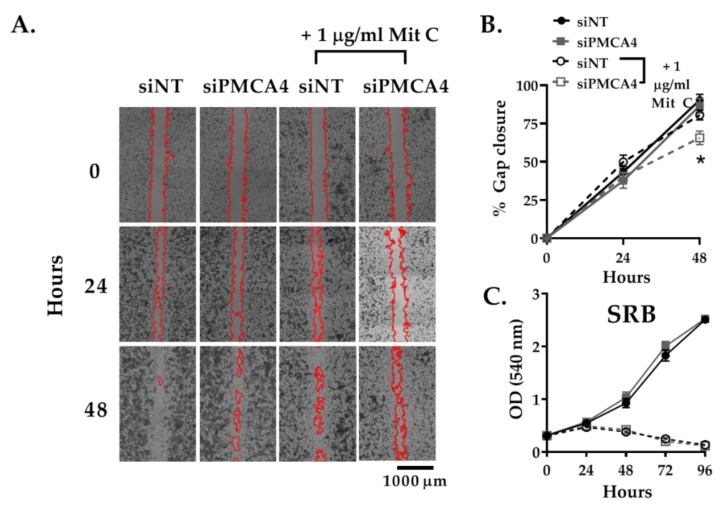
PMCA4 knockdown inhibits cell migration without any effect on cell proliferation. MIA PaCa-2 cells seeded into Ibidi chambers were incubated with non-targeting control (siNT) or siRNA targeting PMCA4 mRNA (siPMCA4). After 24 h, the Ibidi chambers were removed to create a cell-free gap area at time 0. The anti-proliferative agent, 1 μg/ml mitomycin C (Mit C), was added after Ibidi chamber removal to prevent cell proliferation from influencing gap closure. (**A**) Representative images of the gaps at 0, 24, and 48 h. Gap areas (red outline) were processed and analyzed using ImageJ. (**B**) Data are presented as % gap closure with respect to time 0. (**C**) SRB proliferation assays were run in parallel to the migration assay to ensure cell proliferation was sufficiently inhibited by Mit C. Data are shown as mean ± SEM. (n = 4, 3–4 replicates per treatment condition). Comparisons were made between siNT control and siPMCA4 treated cells, with and without Mit C, at matching time points using two-way ANOVA with Tukey’s multiple comparison post-hoc test. * represents a statistically significant difference between % gap closure of siNT + Mit C and siPMCA4 + Mit C, at 48 h, where *p* < 0.05.

**Figure 6 cancers-12-00218-f006:**
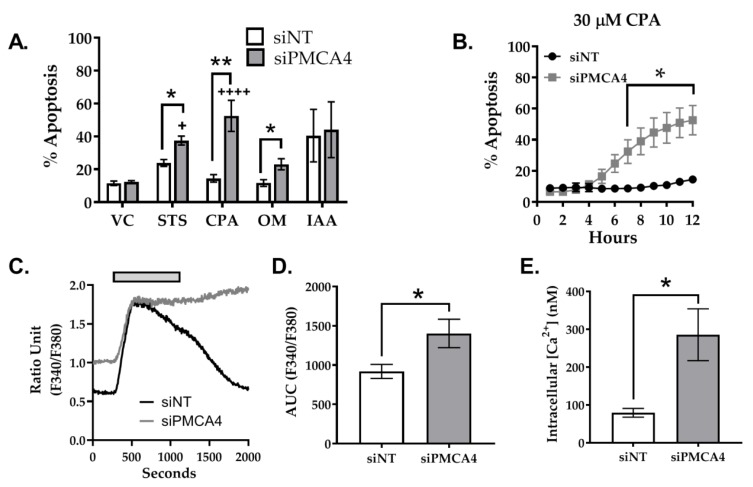
PMCA4 knockdown enhances apoptosis associated with Ca^2+^ overload. MIA PaCa-2 cells were seeded in either non-targeting control (siNT) or siRNA targeting PMCA4 (siPMCA4) for 48 h then treated with either 0.1% DMSO vehicle control (VC), apoptotic-inducing staurosporin (STS), cyclopiazonic acid (CPA), oligomycin (OM) or iodoacetate (IAA). The cells were labelled with Nuclear-ID red stain and caspase3/7 reagent prior to imaging using the Incucyte® live-cell analysis system. (**A**) Cell death is shown as the percentage of caspase 3/7 cleavage with respect to total cell count post-12-hour treatment. (**B**) The effects of CPA-induced apoptosis was further examined at 0.5–12 hr. (**C**) Ca^2+^ overload assay was used to examine the relationship between CPA and Ca^2+^ overload-associated cell death. Fura-2 loaded cells were perfused with HPSS for 200 seconds. After baseline signals were stable, cells were perfused with 30 µM CPA added HPSS (grey box) for 15 minutes and were washed with HPSS for 15 minutes. Representative Ca^2+^ overload traces of siNT control (black) and siPMCA4 (grey) are shown. (**D**) Baseline-subtracted mean area under the curve (AUC) of the uncalibrated fura-2 fluorescence ratios and (**E**) calibrated resting [Ca^2+^]_i_ for siNT vs. siPMCA4 treated cells are shown. Statistical significance comparisons between siNT and siPMCA4 treated conditions were determined by Kolmogorov-Smirnov test and unpaired t-test with Welch’s correction. Data are shown as mean ± SEM. (N = 3–5 experiments; 3 replicates per treatment condition) * and ** represent statistically significant differences between siNT vs. siPMCA4 where *p* < 0.05 and *p* < 0.01, respectively. + and ++++ represent statistically significant differences between VC vs. treatment conditions where *p* < 0.05 and *p* < 0.0001, respectively.

**Figure 7 cancers-12-00218-f007:**
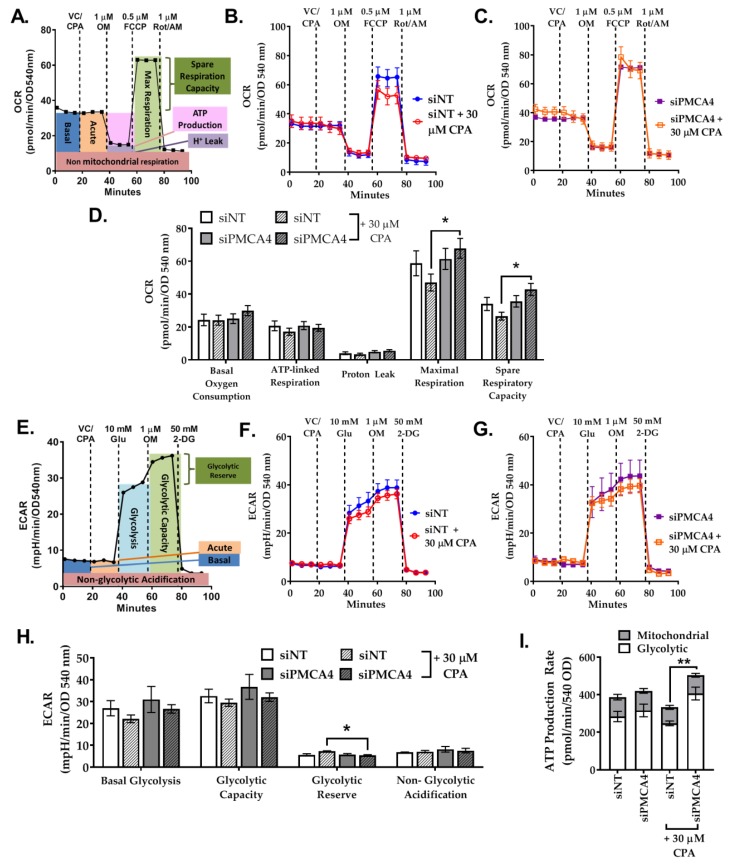
PMCA4 knockdown has minimal effects on the metabolic phenotype. MIA PaCa-2 cells were incubated with either siNT control or siPMCA4 for 48 h. (**A**) Cartoon depicting the experimental design and metabolic parameters measured of Agilent Seahorse XF mito stress test which includes pre-injection treatment (vehicle control: VC, or CPA) and sequential injections of oligomycin (OM), FCCP and a combination of rotenone/antimycin (Rot/AM). Representative traces of oxygen consumption rate (OCR) comparing the effects of VC and 30 μM CPA on (**B**) siNT control and (**C**) siPMCA4. (**D**) Mito stress test data were normalized to protein content (SRB; OD 540 nm). (**E**) Cartoon depicting the experimental design and metabolic parameters measured of Agilent Seahorse XF Glyco stress test which includes pre-injection treatment (vehicle control: VC, or CPA) and sequential injections of D-glucose (Glu), OM and 2-deoxyglucose (2-DG). Representative traces of extracellular acidification rate (ECAR) comparing the effects of VC and 30 μM CPA on (**F**) siNT control and (**G**) siPMCA4 treated cells. (**H**) Glyco stress test data were normalized to protein content. Statistical significance was determined by One-way ANOVA with Tukey multiple comparisons test. (**I**) Agilent Seahorse ATP production rates were derived from mito stress test OCR and ECAR and data are presented as total ATP production rate. * and ** represent a statistically significant difference between siNT + CPA and siPMCA4 + CPA, where *p* < 0.05 and *p* < 0.01, respectively (N = 3–4; 4 replicates per treatment conditions).

**Table 1 cancers-12-00218-t001:** Primer sequences.

Primer ID	Catalogue #	Sequence
ATP2B1_Forward	H_ATP2B1_1	ATCCTCTTGTCTGTAGTGTG
ATP2B1_Reverse	H_ATP2B1_1	TCACCATATTTCACTTGAGC
ATP2B2_Forward	H_ATP2B2_1	GATAGTGATCGTGCAGTTTG
ATP2B2_Reverse	H_ATP2B2_1	AATGAATATGCACCACATCC
ATP2B3_Forward	H_ATP2B3_1	CACCCACTACAAAGAGATTC
ATP2B3_Reverse	H_ATP2B3_1	GTAGTATTTTGGTGGTATAGGC
ATP2B4_Forward	H_ATP2B4_1	AACTCTCAGACTGGAATCATC
ATP2B4_Reverse	H_ATP2B4_1	ACCTTTCTTCTTTTTCTCCC
PMCA4b_Forward	N/A	CCAGACTCAGATCAAAGTGGTCA
PMCA4b_Reverse	N/A	TCGTGGCAACTCCTCCTCTA
S18rRNA_Forward	H_RN18S1_1	ATCGGGGATTGCAATTATTC
S18rRNA _Reverse	H_RN18S1_1	CTCACTAAACCATCCAATCG

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
