# Peer review of "Plasma Membrane Ca^2+^ ATPase Isoform 4 (PMCA4) Has an Important Role in Numerous Hallmarks of Pancreatic Cancer"

_cancers, 2020, doi:10.3390/cancers12010218_

Round 1
Reviewer 1 Report
To the authors
Pishyaporn Sritangos and colleagues report in this manuscript about PMCA4 overexpression in pancreatic ductal adenocarcinoma (PDAC). They provide clinical correlation, in vitro validation on different cell-lines and some functional tests aimed to propose PMCA4 as a therapeutic target in PDAC. The manuscript covers an interesting topic well, nonetheless, there are few sections that deserve to be restructured, in order to achieve the level and comprehensive overview that a journal like Cancers would aim to.
Major points to consider in subsequent versions:
Page 3; figure 1. The authors analyzed ATP2B4 and -2B1 expression in terms of clinical impact. This is fine, as long as the authors point out how the stratified the patients (over the median vs. below the median and/or quartile). Nonetheless, as a prognostic biomarker in terms of both overall and event-free survival, the clinical characteristics of those patients can deeply impact the HR (i.e. nodal status, TNM, grade and histological subtype, R0 or R1 if surgery was performed, neo- or adjuvant therapy received, etc.) We acknowledge that this can be beyond the scope of the manuscript and anyway not possible with a retrospective in silico interrogation. Nonetheless, if co-variates suitable for multivariate statistical analyses are not available this should be mentioned as a study limitation. Indeed those are important information and should be provided in order to propose PMCA4 as a biomarker. Otherwise, this analysis can only be hypothesis-generating.
Page 16; in paragraph 4.13. Statistics the authors state that they employed ANOVA test. This is fine as long as the data analyzed respect a gaussian distribution. If this is the case, this should be stated. Otherwise, a non-parametric test should be performed.
Minor
The manuscript would benefit from a native-speaker revision.
Page 1-2; in the introduction, the authors also briefly summarize calcium dysregulation, PDAC carcinogenesis, and therapeutic combinations. This section can be slightly expanded, comprising novel findings of this topic and the insights about tumor metabolism, leading to the emergence of aberrant signaling pathways as critical factors modulating central metabolic networks that fuel pancreatic tumors. (PMID: 24556680, PMID: 24084722, PMID: 23073473)
Moreover, in the discussion section, the authors recall the same biological landscape mentioned in the introduction. From a preclinical standpoint, several cancers with terribly poor prognosis could benefit from novel insights derived from modern data about fibrosis in PDAC, drug resistance and calcium homeostasis in PDAC (i.e. PMID: 29903994; PMID: 30866547).
Page 8, figure 5 A. How exactly the authors quantify the wound area by ImageJ? Is this done by quantifying the estimated surface? This should be mentioned.
Page 8; in the paragraph ‘2.6. PMCA4 knockdown sensitizes MIAPaCa-2 cells to apoptosis’ and in discussion section: when discussing the paper by the laboratory of Prendergast (reference 48); I think it is important to mention that that particular study refers to APOPTOSIS and the related mechanisms within this particular PDAC model: nonetheless, a tight correlation exists between impaired calcium-glucose metabolism, immune-infiltrate, angiogenesis and cancer progression, and dissemination to distant sites and to nodal compartment. Indeed, CD8+ T cells and immune cells come and go across the permeable capillaries. Because of these intimate interactions, the capacity of dendritic cells and endothelial cells ECs as antigen-presenting cells (APC) can be also discussed, since several examples have been recently published (i.e. PMID: 29247129; PMID: 31277479; PMIDs: 28713824)
Page 13; in the frame of this thinking, in the last discussion paragraph, the concept of apoptotic susceptibility might be slightly expanded to bystander microenvironmental cells. I personally miss some important insights about tumor milieu role in mediating cancer progression, in both solid and hematologic tumors. Indeed, MIAPaCa-2 cells can resemble a cytokine- and cell-adhesion-independent from tumor niche and stromal microenvironment (supported by new vessel formation and cancer proliferation, irrespective of immune-surveillance. Indeed, I personally miss some references that the intimate interaction between endothelial cells, tumor cells, and CD8+ T cells creates a permissive immune microenvironment that allows undisturbed cancer proliferation (PMID: 30546939, 30619378), making them resistant to both conventional and immune-targeting therapeutically strategy. Remarkably, a defective immunosurveillance allows for the persistence and proliferation of MM cells: an immune-microenvironment disease evolution characterized by exhausted CD8+ cells, overexpressing checkpoint molecules such as ctla4 and PD1, in preclinical models offers suitable targets for increased survival in vivo models, as already demonstrated combining Histone Deacetylase Inhibitor, Calcium directed effects and immunotherapy (PMID: 28596940).
Author Response
1) Pishyaporn Sritangos and colleagues report in this manuscript about PMCA4 overexpression in pancreatic ductal adenocarcinoma (PDAC). They provide clinical correlation, in vitro validation on different cell-lines and some functional tests aimed to propose PMCA4 as a therapeutic target in PDAC. The manuscript covers an interesting topic well, nonetheless, there are few sections that deserve to be restructured, in order to achieve the level and comprehensive overview that a journal like Cancers would aim to.
Authors Response: We would like to thank Referee 1 for their constructive and thorough review of our manuscript
Major points to consider in subsequent versions:
2) Page 3; figure 1. The authors analyzed ATP2B4 and -2B1 expression in terms of clinical impact. This is fine, as long as the authors point out how the stratified the patients (over the median vs. below the median and/or quartile). Nonetheless, as a prognostic biomarker in terms of both overall and event-free survival, the clinical characteristics of those patients can deeply impact the HR (i.e. nodal status, TNM, grade and histological subtype, R0 or R1 if surgery was performed, neo- or adjuvant therapy received, etc.) We acknowledge that this can be beyond the scope of the manuscript and anyway not possible with a retrospective in silico interrogation. Nonetheless, if co-variates suitable for multivariate statistical analyses are not available this should be mentioned as a study limitation. Indeed those are important information and should be provided in order to propose PMCA4 as a biomarker. Otherwise, this analysis can only be hypothesis-generating.
Authors Response: Clinical relevance of ATP2B1-4 expression was obtained by data mining gene chip microarray data from PDAC tissue vs normal healthy tissue from the tumour margin of the same patients (Badea et al 2008, Ref 26). Expression was stratified based on the “median centred expression of ATP2B4 and ATP2B1” (Fig. 1A-E; mentioned in section 2.1 Page 3). However, for the Kaplan Meier survival analysis (Fig. 1F & G) “patient survival was stratified into quartiles based around median expression with the high expressing group (>75th percentile) compared to the low expressing group (<25th percentile). Other clinical characteristics (e.g. nodal status, grade, histology, surgery type) were unfortunately not available from either of these studies. Based on the referee’s suggestion, we have amended the manuscript and acknowledge this as a study limitation (Section 2.1, last paragraph Page 3).
3) Page 16; in paragraph 4.13. Statistics the authors state that they employed ANOVA test. This is fine as long as the data analyzed respect a gaussian distribution. If this is the case, this should be stated. Otherwise, a non-parametric test should be performed.
Authors Response: We apologize for not stating in detail that all data sets were initially tested for normality using the Sharpiro-Wilk test before analysing by the appropriate parametric test (Student’s t-test or one way ANOVA with Tukey’s post-hoc test) or non-parametric test (Man Whitney or Kruskal Wallis with Dunn’s multiple comparisons test). Accordingly, we have edited section 4.13 Statistics (Page 18) to include this information.
Minor
The manuscript would benefit from a native-speaker revision.
4) Page 1-2; in the introduction, the authors also briefly summarize calcium dysregulation, PDAC carcinogenesis, and therapeutic combinations. This section can be slightly expanded, comprising novel findings of this topic and the insights about tumor metabolism, leading to the emergence of aberrant signaling pathways as critical factors modulating central metabolic networks that fuel pancreatic tumors. (PMID: 24556680, PMID: 24084722, PMID: 23073473)
Moreover, in the discussion section, the authors recall the same biological landscape mentioned in the introduction. From a preclinical standpoint, several cancers with terribly poor prognosis could benefit from novel insights derived from modern data about fibrosis in PDAC, drug resistance and calcium homeostasis in PDAC (i.e. PMID: 29903994; PMID: 30866547).
Authors Response: We would like to thank Referee 1 for these insightful comments and have therefore edited the Introduction to include some of these suggestions (see Introduction page 4-5???): “The calcium signalling machinery represents a rich tapestry of potential therapeutic targets. This is because the spatiotemporal shaping of calcium signals are critical for numerous physiological repsponses that are central to cancer hallmark respsonses, including cell proliferation/cell cycle control, apoptosis, migration/invasion, angiogenesis, metabolism (PMID: 28386091) as well as fibrosis and drug resistance that can contribute to poor patient prognosis (PMID: 29903994; PMID: 30866547) It is therefore not surprising that remodelling of the calcium signalling machinery is reported to lead to numerous cancer hallmark responses in numerous different cancers (PMID: 28386091). However, due to the ubiquitous nature of most Ca2+ transport pathways, targeting them with specific drugs will most likely lead to adverse effects, unless they are either uniquely overexpressed or exhibit a unique function. The current study identified PMCA4 as a potential candidate as it appears to be almost uniquely overexpressed, is uniquely coupled to glycolytic metabolic enzymes and has an important role in cancer hallmark responses, including migration and apoptosis resistance. These comments have been addressed in the Introduction (page 2) and Discussion (page 14-15).
5) Page 8, figure 5 A. How exactly the authors quantify the wound area by ImageJ? Is this done by quantifying the estimated surface? This should be mentioned.
Authors Response: The cell-free gap area was quantified using Fiji ImageJ after image pixel calibration. The known pixel size using the 4x lens, captured by the Hanamatsu Orca camera is 1.6125 μm/ 1 pixel. All captured images were “Set Scale” in Fiji ImageJ and the cell-free binary area was measured as the cell-free gap area (μm2). These additional details have now been added to the “Supplementary Methods” (Methods S4) for clarity.
6) Page 8; in the paragraph ‘2.6. PMCA4 knockdown sensitizes MIAPaCa-2 cells to apoptosis’ and in discussion section: when discussing the paper by the laboratory of Prendergast (reference 48); I think it is important to mention that that particular study refers to APOPTOSIS and the related mechanisms within this particular PDAC model: nonetheless, a tight correlation exists between impaired calcium-glucose metabolism, immune-infiltrate, angiogenesis and cancer progression, and dissemination to distant sites and to nodal compartment. Indeed, CD8+ T cells and immune cells come and go across the permeable capillaries. Because of these intimate interactions, the capacity of dendritic cells and endothelial cells ECs as antigen-presenting cells (APC) can be also discussed, since several examples have been recently published (i.e. PMID: 29247129; PMID: 31277479; PMIDs: 28713824)
Authors Response: It’s important to note that there was no discussion of any paper by Prendergast. However, there were mistakes with some of the references cited in the original manuscript due to the reference list being slightly out of sequence which may have caused confusion. When describing Ref 48 (Peters AA (2016) Sci Rep.;6(1):25505) in the Discussion this should have been Eaddy AC et al 2011 (Ref 56 in the original). This relates to the observation that the Ca2+ stress (CPA)-induced reduction in mitochondrial respiration in PMCA4 expressing cells may be due to mitochondrial depolarization. However, when describing the effect of PMCA4 knockdown on either cell proliferation or apoptosis, we cited studies in breast cancer cells from the laboratory of Greg Monteith which were broadly similar to our data. For example, silencing of PMCA2 reduced MDA-MB-231 breast cancer cell proliferation, whereas silencing of the related PMCA4 had no effect, consistent with our study in PDAC cells (Peters AA (2016) Sci Rep.;6(1):25505). Similarly, silencing PMCA4 sensitized breast cancer cells to apoptosis, consistent with our study in PDAC cells, whereas silencing PMCA1 sensitized breast cancer cells to necrotic cell death (Curry et al (2012) J Biol Chem. 287(34):28598–608). We apologise for any confusion that this error may have created. The reference list has now been corrected and the relevant papers are now cited appropriately. The specific comment relating to the links between Ca2+ metabolism with immune infiltration, angiogenesis, etc. will be dealt with in the next comment.
7) Page 13; in the frame of this thinking, in the last discussion paragraph, the concept of apoptotic susceptibility might be slightly expanded to bystander microenvironmental cells. I personally miss some important insights about tumor milieu role in mediating cancer progression, in both solid and hematologic tumors. Indeed, MIAPaCa-2 cells can resemble a cytokine- and cell-adhesion-independent from tumor niche and stromal microenvironment (supported by new vessel formation and cancer proliferation, irrespective of immune-surveillance. Indeed, I personally miss some references that the intimate interaction between endothelial cells, tumor cells, and CD8+ T cells creates a permissive immune microenvironment that allows undisturbed cancer proliferation (PMID: 30546939, 30619378), making them resistant to both conventional and immune-targeting therapeutically strategy. Remarkably, a defective immunosurveillance allows for the persistence and proliferation of MM cells: an immune-microenvironment disease evolution characterized by exhausted CD8+ cells, overexpressing checkpoint molecules such as ctla4 and PD1, in preclinical models offers suitable targets for increased survival in vivo models, as already demonstrated combining Histone Deacetylase Inhibitor, Calcium directed effects and immunotherapy (PMID: 28596940).
Authors Response: We would like to thank Referee 1 for some insightful comments. We would agree that in the context of the glycolytic phenotype, PMCA4 function and apoptosis resistance/sensitization, the tumour microenvironment may play an important role in facilitating tumour progression that needs to be addressed in the revised manuscript. Specifically, the tumour microenvironment is notoriously highly acidic, due to the highly glycolytic phenotype and lactic acid efflux (PMID: 16707446). This acidic microenvironment facilitates matrix metalloproteinases and thus migration and invasion (PMID: 24204192[PS1] ), but also inhibits immune surveillance (PMID: 28267587). Specifically, tumor acidity acts as a broad immune escape mechanism by which cancer cells, inhibit antitumour immune effectors (including T cells, NK cells and crucial antigen-presenting dendritic cells), while simultaneously promoting the immunosuppressive and protumour properties of regulatory T cells and myeloid cells. However, in the context of the current study, the acidic microenvironment may also specifically promote PMCA4 activity and thus further promote cell survival. This is because the PMCA is an ATP-driven Ca2+/H+ exchanger and thus extracellular acidification facilitates Ca2+ efflux and alkalinisation inhibits Ca2+ efflux (PMID: 2156439; PMID: 17485401). This means that a highly glycolytic PDAC cell phenotype not only provides a privileged ATP supply to the PMCA but also further accentuates PMCA activity by promoting lactic acid efflux and extracellular acidification. This therefore means that within the context of a hypoxic acidic tumour microenvironment in vivo, over-expression of PMCA4 may have an even more pro-survival, promigratory phenotype than observed in our 2D cell culture model. This also suggests that targeting PMCA4 using novel and specific inhibitors may be especially effective in combination with drugs that target tumour acidification. These include glycolytic inhibitors (PKM2, PMID: 31819190), lactic acid/monocarboxylate transporter (MCT) inihibitors (PMID: 31395464), lactate dehydrogenase (LDH) inhibitor (PMID: 29120638; PMID: 30248111) and hypoxia inducible factor (HIF1alpha) (PMID: 27139518). In addition, these drugs may also have a synergistic effect when combined with other novel immunotherapies such as checkpoint inhibitors (PMID: 31577944). These points have now been expanded in the Discussion (page 14-15).
Reviewer 2 Report
A well conducted and comprehensive study on the role of plasma membrane calcium ATPase-4 in pancreatic adenocarcinoma biology.
Author Response
Authors Response: We would like to thank Referee 2 for the highly positive comments and are pleased that he/she found merits in the manuscript.
Reviewer 3 Report
This study about PMCA4 in pancreatic cancer, in the opinion of Authors, demonstrates that this molecule offer an attractive model for therpeutic strategies against pancreatic cancer.
However, it is very well known that the Literature is full of manuscripts like this with potential interesting findings but without a real translational potential.
This study, in particular, does not show any experiments in more reliable models (in vivo models, mice, organoids) and its real clinical value is very far to be important or to be translated in the clinics.The calcium-related signaling in PDAC is also very complex, but the Authors did not try to integrate their results into this already well known signaling-metabolism and molecular pathways.
Although I acknowledge some merits to this investigation, it appears to me a paper with a very low priority in the landscape of pancreatic cancer, perhaps more suitable to a "basic science" Journal (although it remains also in that field a very low priority study in my opinion).
Author Response
This study about PMCA4 in pancreatic cancer, in the opinion of Authors, demonstrates that this molecule offer an attractive model for therpeutic strategies against pancreatic cancer.
However, it is very well known that the Literature is full of manuscripts like this with potential interesting findings but without a real translational potential.
This study, in particular, does not show any experiments in more reliable models (in vivo models, mice, organoids) and its real clinical value is very far to be important or to be translated in the clinics.The calcium-related signaling in PDAC is also very complex, but the Authors did not try to integrate their results into this already well known signaling-metabolism and molecular pathways.
Although I acknowledge some merits to this investigation, it appears to me a paper with a very low priority in the landscape of pancreatic cancer, perhaps more suitable to a "basic science" Journal (although it remains also in that field a very low priority study in my opinion).
Authors Response: We would like to thank Referee 3 for the comments and feedback and are grateful that he/she found some merits in our study. We also understand and are very sympathetic with the view that our study lacks clinical translation and might only add an incremental step to the dearth of information and studies claiming to have discovered a novel therapeutic target without ever reaching the clinic. We respectfully disagree and would like to re-emphasise some key scientific arguments that might quell some of these concerns. We also feel that our study is timely and warrants early publication as it provides a foundation and springboard so that other researchers, in addition to ourselves, can build upon this knowledge to speed up the clinical translation.
Lack of clinical translation: Although our studies have not yet been translated to animal models, which we argue might not provide sufficient mechanistic insight in a timely manner (see point below: Lack of animal models), the data mining (Fig 1) provides an important translational link to patient survival. Most notably these data suggest that PMCA4 is almost exclusively overexpressed in human PDAC tumours, compared to other PMCA isoforms, and that this leads to poor patient survival. This is an important aspect of our study that provides a strong rationale for dissecting the specific mechanism for the role of PMCA4 in PDAC cells, which can only really be achieved using the more reductionist cellular model, MIA PaCa-2 cells. The strength of using MIA PaCa-2 cells is that they recapitulate important features of PDAC cells in vivo; almost exclusive expression of PMCA4, high glycolytic metabolism, paid cell growth as well as the essential expression of mutant KRas and p53.
PMCA4 as a novel bona fide therapeutic target: “The calcium signalling machinery represents a rich tapestry of potential therapeutic targets. This is because the spatiotemporal shaping of calcium signals is critical for numerous physiological responses that are central to cancer hallmark responses, including cell proliferation/cell cycle control, apoptosis, migration/invasion, angiogenesis (PMID: 28386091) as well as fibrosis and drug resistance that can contribute to poor patient prognosis (PMID: 29903994; PMID: 30866547) It is therefore not surprising that remodelling of the calcium signalling machinery is reported to lead to numerous cancer hallmark responses in numerous different cancers (PMID: 28386091). However, due to the ubiquitous nature of most Ca2+ transport pathways, targeting them with specific drugs will most likely lead to adverse effects, unless they are either uniquely overexpressed in the tumour or exhibit a unique function. The current study identified PMCA4 as a potential candidate as it appears to be almost uniquely overexpressed in PDAC tissue and PDAC cells, and is also functionally coupled to the glycolytic enzyme machinery (James et al 2019; PMID ). This is particularly important and lies at the crux of the argument that PMCA4 might represent a bona fide therapeutic target. This is because PMCA4 knockout mice exhibit almost no adverse phenotype, presumably because most tissues express other PMCA isoforms, particularly the ubiquitous PMCA1, thereby resulting in functional redundancy. The only cell type that exhibits an adverse phenotype is sperm that exclusively expresses PMCA4 and consequently PMCA4 knockout male mice are infertile. Extrapolating this to PDAC cells which also appear to almost exclusively PMCA4 suggests that targeting PMCA4 by acute treatment with potent and selective drugs might be particularly effective at inducing Ca2+ stress induced apoptosis, when combined with other Ca2+ stressors. These points have now been emphasised in the Introduction (page 3) and Discussion (page 14-15) of the revised manuscript.
Lack of animal models: There are numerous scientific and logistical reasons why translating our study to animal models would not only markedly delay publication of this timely study, but not necessarily provide us with much additional mechanistic insight at this early stage. Moreover, the use of any animal models requires extensive ethical and Home Office approval that is additionally time consuming. The simplest model would be a xenograft model in which PMCA4 is either stably knocked down or over-expressed in PDAC cells prior to implantation into the xenograft model. However, stable knockdown and over-expression of PMCA4 is not trivial, is very time consuming and requires extensive optimization. Moreover, based on our cell proliferation assays, it’s unlikely that this would affect tumour growth, which is the main functional readout in any xenograft model. Therefore, we are left with testing the effect of PMCA4 expression (knockdown/over-expression) on cell death, but would require testing the effect of numerous Ca2+ stressor agents to determine whether this is potentiated (PMCA4 knock down) or protected (PMCA4 over-expression). This may introduce numerous adverse effects, confounding factors and ethical considerations for the mice. There are also concerns that a xenograft model does not accurately reflect the tumour microenvironment or pathological progression of PDAC in the human disease. The genetically modified mouse models, such as the KPC mouse much better reflects the human disease. However, this would need to be crossed with a conditional PMCA4 knockout/over-expressing mouse, which is both technically demanding, expensive and even more time consuming. Therefore on balance translating to animal models is technically and financially unfeasible given the limited time allowed by the journal (10 days) to address these issues.
Integration of our results into the complex Ca signalling-metabolism-molecular pathways: On reflection we agree with Referee 3 that we need to place or results into context with prominent and emerging themes of metabolic remodelling, the tumour microenvironment and immune surveillance in PDAC. We have now provided a detailed description of how the acidic tumour microenvironment provides an important contextual in vivo link between glycolytic phenotype, PMCA4 over-expression and function, matrix remodelling and immune surveillance (see Discussion, page 14-15). This is because the acidic microenvironment facilitates matrix metalloproteinases (PMID: and thus migration and invasion (PMID 24204192), but also inhibits immune surveillance (PMID: 28267587). Specifically, tumor acidity acts as a broad immune escape mechanism by which cancer cells, inhibit antitumour immune effectors (including T cells, NK cells and crucial antigen-presenting dendritic cells), while simultaneously promoting the immunosuppressive and protumour properties of regulatory T cells and myeloid cells. However, in the context of the current study, the acidic microenvironment may also specifically promote PMCA4 activity and thus further promote cell survival. This is because the PMCA is an ATP-driven Ca2+/H+ exchanger and thus extracellular acidification facilitates Ca2+ efflux and alkalinisation inhibits Ca2+ efflux (PMID: 2156439; PMID: 17485401). This means that a highly glycolytic PDAC cell phenotype not only provides a privileged ATP supply to the PMCA but also further accentuates PMCA activity by promoting lactic acid efflux and extracellular acidification. This therefore means that within the context of a hypoxic acidic tumour microenvironment in vivo, over-expression of PMCA4 may have an even more pro-survival, promigratory phenotype than observed in our 2D cell culture model. This also suggests that targeting PMCA4 using novel and specific inhibitors may be especially effective in combination with drugs that target tumour acidification. These include glycolytic inhibitors (PKM2, PMID: 31819190), lactic acid/monocarboxylate transporter (MCT) inhibitors (PMID: 31395464), lactate dehydrogenase (LDH) inhibitor (PMID: 29120638; PMID: 30248111) and hypoxia inducible factor (HIF1alpha) (PMID: 27139518). In addition these drugs may also have a synergistic effect when combined with other novel immunotherapies such as checkpoint inhibitors (PMID: 31577944). These points have now been expanded in the Introduction (page 3).
Reviewer 4 Report
Well written and presented manuscript. The work is novel as there is no existing literature on the role of PMCA4 in pancreatic cancer.
The manuscript demonstrates that plasma membrane Ca2+ ATPase isoform 4 (PMCA4), that facilitates calcium signaling, plays an important role in pancreatic cancer. The authors have utilized two open data source to determine the changes in PMAC1-4 family members in pancreatic cancer and their correlation with poor patient survival. The authors further conducted studies in cell culture and determined that PMCA4 is highly over-expressed in pancreatic ductal carcinoma cells and plays a key role in migration and apoptosis resistance in these cells. Overall, the studies are very interesting, the experiments are well thought and performed and the results support the conclusion. The manuscript is well written and presented. A minor suggestion is to provide more detail of the pancreatic cell lines used in the study.
Author Response
The manuscript demonstrates that plasma membrane Ca2+ ATPase isoform 4 (PMCA4), that facilitates calcium signaling, plays an important role in pancreatic cancer. The authors have utilized two open data source to determine the changes in PMAC1-4 family members in pancreatic cancer and their correlation with poor patient survival. The authors further conducted studies in cell culture and determined that PMCA4 is highly over-expressed in pancreatic ductal carcinoma cells and plays a key role in migration and apoptosis resistance in these cells. Overall, the studies are very interesting, the experiments are well thought and performed and the results support the conclusion. The manuscript is well written and presented. A minor suggestion is to provide more detail of the pancreatic cell lines used in the study.
Authors Response: We would like to thank Referee 4 for these positive comments. We have now included additional details of the different pancreatic cell lines used in this study (section 2.2; first paragraph, page 4-5). We added information about the origin of the cells and cited the papers that established these cell cultures.
Reviewer 5 Report
Manuscript ID: cancers-654808
Pancreatic ductal adenocarcinoma (PDAC) is a major burden on human health. While in some other cancer types targeted therapy (for example with tyrosine kinase inhibitors) or immunotherapy (for example using immune checkpoint inhibitors) can lead to promising, and sometimes very significant antitumor responses, currently available therapy of PDAC is not efficient. It is clear, that new approaches for the successful treatment of PDAC need to be developed.
The laboratory of the Authors explores this issue since several years by investigating calcium transport processes in PDAC cell lines. Because the perturbation of cellular calcium homeostasis can lead to cell death, and because plasma membrane calcium pumps (PMCA enzymes) play an essential role in the control of cytosolic calcium levels, it is proposed that a better understanding of PMCA function in PDAC may help to identify previously unknown types of vulnerabilities in PDAC cells that may be leveraged for therapy. A strong inhibition of PMCA-dependent calcium transport selectively targeted to PDAC cells in a patient would almost certainly lead to very significant antitumor effects.
PMCA enzymes use the energy of ATP hydrolysis for calcium transport. Data in the literature, including those of the laboratory of the Authors indicate that in PDAC cells PMCA enzymes obtain ATP from glycolysis, probably taking place in the proximity of the intracellular face of plasma membrane. This local glycolytic ATP production (related in tumors to the Warburg effect) may constitute a privileged energy source for plasma membrane located ATPases such as PMCA enzymes.
In the present work Authors investigated the expression of PMCA isoenzymes in human pancreatic cancers, in normal primary pancreatic cells (ductal and stellate) and in two pancreatic cancer cell lines (MIA PaCa-2 and PANC-1) grown in vitro. Authors identify PMCA4 as the main PMCA enzyme expressed in MIA PaCa-2 cells, and, using a PMCA4-specific knockdown approach, they show that PMCA4 is important for MIA PaCa-2 cell migration, that PMCA4 knockdown leads to decreased calcium extrusion from the cells and sensitizes the cells to calcium overload-related apoptosis. Authors also show that in these cells, ATP is synthesized mainly by glycolysis, and that PMCA4 knockdown exerts modest effects on cell energy metabolism. Exploration of PMCA4 function and related metabolic effects in these cells contributes to a better understanding of PDAC biology.
The paper is clearly written, data are presented in a comprehensible manner, and statistical analysis is included.
Comments:
1. References N° 53 and 54 are the same. Later References will need to be renumbered.
2. Please replace “Ca2+” by “Ca2+“ in References.
3. Refs. N° 25 and 27 are not adequate. Give URL (in the Manuscript), as well as more detailed information regarding searches (in the Supplemental section) in order to enable the reader to repeat searches.
4. There is an unexplained star sign in Ref. N° 50.
5. P.2/20, lanes 48-49: “There are four isoforms of PMCAs (PMCA1-4) encoded by four distinct genes (ATP2B1-4).”: This statement is somewhat problematic, because in reality the four PMCA genes (ATP2B1, 2, 3 and 4) give rise by alternative splicing to more than twenty protein isoforms, each single PMCA gene coding for several alternative splice isoforms. This is a relevant issue in this paper, for example because some PMCA4-specific primers used are actually PMCA4b-specific (lanes 488-489).
6. As positive control (Fig. 2, Panel A) for the expression of PMCA 1, 2, 3 and 4-type PMCA enzymes mouse brain lysates were used. Probably it would have been more appropriate to use a cell line individually transfected with the four respective human PMCA cDNAs.
7. Authors compare PDAC PMCA expression to normal adjacent pancreatic tissue, as well as to pancreatic ductal epithelial cells and pancreatic stellate cells grown in vitro. It is not entirely clear to the Reviewer, what is the exact relevance of these comparisons? PDAC tissue contains variable, and sometimes quite substantial amounts of stroma and associated mesenchymal cell types. In addition, normal pancreatic tissue is mostly acinar epithelium, and can contain islets of Langerhans with their own various cell types. The purpose of these comparisons and the interpretation of the observed differences should be stated more clearly. Comparative PMCA4 immunohistochemistry would be probably more informative.
8. Survival on Kaplan-Meyer plots as a function of PMCA4 expression doesn’t seem to present significant differences until approximately 600-700 days post-diagnosis; the curves really start to diverge only thereafter. Could the Authors propose an interpretation for this, and could they analyze the two parts of the curves separately?
9. MIA PaCa-2 cells contain much more PMCA4 than PANC-1 cells (Fig. 2, Panel A). It is therefore not justified to extrapolate results obtained on MIA PaCa-2 cells to every PDAC. This should be mentioned. Maybe poorly differentiated pancreatic ductal adenocarcinoma (the histological type of the original tumors from which both cell lines were established) could be stratified according to PMCA4 expression. It is also worth mentioning, that the majority of PDACs is well or moderately differentiated histologically, rather than poorly differentiated; this may influence PMCA4 abundance. Investigation of additional PDAC cell lines would be interesting. Of course, these issues do not compromise the validity of observations made on MIA PaCa-2 cells; however, they need to be integrated into the Discussion.
10. P. 12/20, lanes 373-374: “We then identified MIAPaCa-2 cell line as an ideal cellular model of PDAC based on the almost exclusive expression of PMCA4.” Considering that PANC-1 cells are also a PDAC cell line, which, however, express much less PMCA4, this phrase is somewhat confusing. In fact, MIA PaCa-2 may constitute a well-suited model for an eventual “PMCA4 high” molecular subtype of PDAC. This issue could be explored by various types of genomic and transcriptomic analyses.
11. P. 14/20, lane 459: Details and references for mycoplasma detection should be added. Presumably, contaminated cultures were not used.
12. What was the estimated purity (%) of the primary pancreatic ductal and stellate cell cultures used in this study, and how was this measured?
13. P. 15/20, lanes 501-513: How was equal loading of samples controlled? Were protein concentration measurements of lysates and, for example Ponceau red staining and densitometry of lanes on blots done, or does control of equal loading rely exclusively on b-actin?
14. P. 7/20, lane 206: To obtain calcium influx following cyclopiazonic acid treatment, 20 mM Ca2+ is added to obtain store operated calcium entry into the cells. This seems to be a rather supra-physiological concentration. What motivated the use of 20 mM Ca2+?
15. Authors discuss cyclopiazonic acid-induced effects in the context of cytoplasmic calcium overload induced by store operated calcium entry. However, it should be kept in mind, that cyclopiazonic acid treatment can induce cell stress also due to endoplasmic reticulum calcium depletion (“endoplasmic reticulum stress”).
16. Knock down of PMCA4 in this work was partial, reaching an approximately 70% down-regulation of expression, and the resting cytosolic calcium levels of PMCA4 knock-down cells was strongly elevated when compared to control cells (P.9/20, lane 281). Because the calcium transport activity of PMCA enzymes is strongly enhanced by calcium/calmodulin and other mechanisms, it should be noted in the manuscript, that the remaining (30%) PMCA4 enzyme, as well as PMCA1, also present in the cells, were very probably in a strongly stimulated state, and that their stimulated transport activity contributed significantly to the re-establishment of the calcium homeostasis (and hence survival) of the cells after PMCA4 knockdown.
17. Please give more detailed information regarding the PMCA2 and 3-specific antibodies used in this work.
18. Authors state that “targeting PMCA4 may potentially be beneficial as a therapeutic adjuvant which selectively sensitizes PDAC cells to currently available clinical therapy while sparing healthy tissues” (p. 13/20, lanes 448-449). How could this be accomplished in a clinically meaningful manner?
19. S2, Panel A (Legend): “Representative TMRE traces are shown with black arrows…” All arrows in this Panel are black (delete "black").
20. Fig. S5 and Legend are somewhat confusing, and colors in this Figure are hard to distinguish. Calculation of R2/R1 should be explained more clearly.
21. Ref. 34 (lanes 692-693) deals with programmed cell death induced by endoplasmic reticulum stress in soybean cells. Is this an adequate reference as used in lane 389 of the Manuscript, for Jurkat cells?
22. Please note that the correct spelling of the PDAC line used in this study as per ATCC or the original paper of Yunis et al., Int. J. Cancer, 1977, is:
MIA PaCa-2.
Author Response
Authors Response: We would like to thank Referee 5 for these very positive comments and constructive feedback. We will attempt to deal with some of the more specific details on a point-by-point basis below.
Comments:
1. References N° 53 and 54 are the same. Later References will need to be renumbered.
Authors Response: We thank the referee for pointing out this error. The same references (previously No. 53-54 and is currently ref no. 60) has been corrected. All subsequent references have been corrected and renumbered
2. Please replace “Ca2+” by “Ca2+“ in References
Authors Response: We thank the referee for pointing out this error. All superscript in the references have been corrected.
3. Refs. N° 25 and 27 are not adequate. Give URL (in the Manuscript), as well as more detailed information regarding searches (in the Supplemental section) in order to enable the reader to repeat searches.
Authors Response: As the referee has suggested, an additional explanation of specific search terms and search sequences (Method S1) has been added to the supplementary materials to facilitate repeated searches. The URLs for both Oncomine website and the search result for The Human Protein Atlas databases were available as part of the reference section. The Oncomine specific search URL (https://www.oncomine.org/resource/main.html#d%3A149671257%3Bdso%3AgeneOverex%3Bdt%3Adataset%3Bec%3A%5B2%5D%3Bepv%3A150001.151078%2C2937%2C3508%3Bet%3Aover%3Bf%3A1900610%3Bg%3A493%3Bp%3A149672127%3Bpg%3A1%3Bpvf%3A3471%2C5448%2C150004%3Bscr%3Adatasets%3Bss%3Aall%3Bv%3A18) unfortunately requires subscription and has limited functionality otherwise. Therefore, a new search session through the main www.oncogene.org is recommended.
4. There is an unexplained star sign in Ref. N° 50.
Authors Response: We apologise for the star sign/emoticon in the reference (previously No 50 and is currently ref no. 57). The star symbol has been removed.
5. P.2/20, lanes 48-49: “There are four isoforms of PMCAs (PMCA1-4) encoded by four distinct genes (ATP2B1-4).”: This statement is somewhat problematic, because in reality the four PMCA genes (ATP2B1, 2, 3 and 4) give rise by alternative splicing to more than twenty protein isoforms, each single PMCA gene coding for several alternative splice isoforms. This is a relevant issue in this paper, for example because some PMCA4-specific primers used are actually PMCA4b-specific (lanes 488-489).
Authors Response: We agree that the PMCA genes (ATP2B1-4) can give rise to alternative splicing, yielding different “splice variants” include the C-terminus a, b, d as well as the N-terminus x and z splice variants (PMID: 24448801). The current work intended to broadly examine ATP2B4 expression and knockdown in our PDAC model. Therefore, pre-design ATPB2B4 primers which broadly detected ATPB2B4 mRNA variants were purchased from sigma Aldrich (catalogue number: H_ATP2B4_1).
However, special emphasis on PMCA4b expression after siPMCA4 knockdown was performed because PMCA4b is reported to be involved in multiple pro-proliferative signalling mechanisms (e.g. growth and cell cycle) which potentially associates with cancer. Therefore, we anticipated that if ATP2B4b splice variant expression was sufficiently silenced, this may inhibit cell proliferation and the pro-proliferative signal transduction pathways. However, to our surprise, although ATP2B4 expression (including ATP2B4b) was knocked down sufficiently to impair PMCA-mediated Ca2+ clearance and homeostasis, we did not observe any effect on Ca2+ overload-independent PDAC cell viability nor cell proliferation rate.
6. As positive control (Fig. 2, Panel A) for the expression of PMCA 1, 2, 3 and 4-type PMCA enzymes mouse brain lysates were used. Probably it would have been more appropriate to use a cell line individually transfected with the four respective human PMCA cDNAs.
Authors Response: We would like to thank Referee 5 for these insightful comments Brain lysates were used as a positive control as they are a rich source of all 4 PMCA isoforms and thus useful to confirm that each antibody works. We concur that it would be very useful to include additional controls to test the specificity of each PMCA-specific antibody using cell lines in which each specific PMCA-isoform is over-expressed exclusively. However, this is not trivial and would require extensive optimization and multiple steps including the stable over-expression of the specific PMCA isoform followed by sequential stable deletion of all remaining endogenous PMCA isoform expression. This is the focus of our ongoing work but is very time consuming, especially given that over-expression of PMCA1 for example is notoriously difficult. Therefore, we anticipate that this would take several months to achieve and is very expensive and would therefore not necessarily justify the potential added value to the study above and beyond using brain lysates.
7. Authors compare PDAC PMCA expression to normal adjacent pancreatic tissue, as well as to pancreatic ductal epithelial cells and pancreatic stellate cells grown in vitro. It is not entirely clear to the Reviewer, what is the exact relevance of these comparisons? PDAC tissue contains variable, and sometimes quite substantial amounts of stroma and associated mesenchymal cell types. In addition, normal pancreatic tissue is mostly acinar epithelium, and can contain islets of Langerhans with their own various cell types. The purpose of these comparisons and the interpretation of the observed differences should be stated more clearly. Comparative PMCA4 immunohistochemistry would be probably more informative.
Authors Response: We have edited section 2.2 (1st paragraph, page 4-5) in an attempt to clarify the purpose of these comparisons. We also agree with the referee that the PDAC tumour and resected tissue may contain variable types of cells. The comparison between PDAC PMCA expression vs the healthy tissue from the tumour margin was performed through data mining to gauge the clinical relevance of PMCA expression in PDAC disease. The key purposes of this comparison are the following: 1) to broadly identify any altered PMCA1-4 expression in PDAC tumour tissue compared to healthy tissue, and 2) to examine whether these specific PMCA isoforms expression could be correlated to PDAC patient survival. As it turned out, PMCA4 is overexpressed and could be correlated to poor PDAC patient survival while PMCA1 overexpression could not be correlated to PDAC patient survival (Figure 1). Therefore, we shifted our focus onto PMCA4 overexpression and attempted to identify a cellular PDAC model which reflects this characteristic.
In order to identify such a cellular model, in Figure 2, we compared two PDAC cell lines (MIA PaCa-2 and Panc-1) and two non-cancer/non-malignant pancreatic cells (hPSC and HPDE). Except for the MIA PaCa-2 cells which had high ATP2B4:ATP2B1 ratio, other cell lines had relatively similar ATP2B4:ATP2B1 expression ratios. Furthermore, the Western blot data suggest that MIA PaCa-2 express more PMCA4 protein compared to other cell lines. Interestingly, our previous studies (James, AD et al 2013 and 2015 (PMID 26294767 and 24158437) suggested that PMCA are reliant on glycolysis in MIA PaCa-2 cells and thus identified these cells as being more reliant on glycolytically –derived ATP than the related Panc-1 cells. As our ongoing research is interested in the relationship between PMCA and glycolysis, this makes MIA PaCa-2 an ideal cellular model to examine the role of PMCA4 on PDAC cancer hallmarks and particularly the glycolytic shift towards glycolysis. We have tried to convey this additional information in the revised manuscript (Section 2.2. page 5).
8. Survival on Kaplan-Meyer plots as a function of PMCA4 expression doesn’t seem to present significant differences until approximately 600-700 days post-diagnosis; the curves really start to diverge only thereafter. Could the Authors propose an interpretation for this, and could they analyze the two parts of the curves separately?
Authors Response: The authors agree with the referee’s observation that the curve does not diverge very soon after diagnosis. Unfortunately, as these data were mined from a previous study, we are unsure of what factors might contribute to this observation. Current evidence in a breast cancer model suggests that PMCA4 expression plays a role in apoptotic resistance. Thereby, we can only hypothesize that PMCA4 overexpression potentially benefits PDAC tumour survival and hindering therapeutic treatment, leading to poor patient prognosis and overall survival.
It should be noted that Kaplan-Meier survival curves presented in this study were analysed using log-rank test. This is the statistical test normally used to assess survival data and generally assess the survival of the patient cohort over a period of time. The significance (“*”) on the ATP2B4 Kaplan-Meier curve represents the significance of the entire curve rather than any given single time point. As the Kaplan-Meier curve shows the loss of patient from the cohort at various time points, no individual parts of the curve were analysed separately in this study. Therefore, the authors apologize for any misunderstanding. We have attempted to clarify this significance by stating in figure 1 legend text that the entire survival curve was used for statistical comparison using the log-rank test.
9. MIA PaCa-2 cells contain much more PMCA4 than PANC-1 cells (Fig. 2, Panel A). It is therefore not justified to extrapolate results obtained on MIA PaCa-2 cells to every PDAC. This should be mentioned. Maybe poorly differentiated pancreatic ductal adenocarcinoma (the histological type of the original tumors from which both cell lines were established) could be stratified according to PMCA4 expression. It is also worth mentioning, that the majority of PDACs is well or moderately differentiated histologically, rather than poorly differentiated; this may influence PMCA4 abundance. Investigation of additional PDAC cell lines would be interesting. Of course, these issues do not compromise the validity of observations made on MIA PaCa-2 cells; however, they need to be integrated into the Discussion.
Authors Response: We thank the referee for this insight. It is very interesting that PMCA4 expression is reported to be associated with cell differentiation (e.g. colon cancer) and is dependent on cell type (e.g. sperm).Based on the literature, we find that both MIA PaCa-2 and PANC-1 are derived from primary PDAC tumour and are both reported to be poorly differentiated PDAC cell lines (reviewed by PMID: 20418756). Indeed, It would be extremely interesting to investigate PMCA4 expression stratification using a larger panel of different pancreatic cell lines. We have tried to convey the caveat in the Discussion that extrapolation of PMCA4 expression is not universal and is restricted to high-PMCA4 expressing subtypes in the discussion (paragraph 3).
10. P. 12/20, lanes 373-374: “We then identified MIAPaCa-2 cell line as an ideal cellular model of PDAC based on the almost exclusive expression of PMCA4.” Considering that PANC-1 cells are also a PDAC cell line, which, however, express much less PMCA4, this phrase is somewhat confusing. In fact, MIA PaCa-2 may constitute a well-suited model for an eventual “PMCA4 high” molecular subtype of PDAC. This issue could be explored by various types of genomic and transcriptomic analyses.
Authors Response: We have edited these sentences to clarify that we primarily focused on “PMCA4-expressing PDAC” subtype.
11. P. 14/20, lane 459: Details and references for mycoplasma detection should be added. Presumably, contaminated cultures were not used.
Authors Response: We have amended the manuscript (section 4.1 Cell culture, page 15) to include more details regarding mycoplasma test. Of course contaminated cultures were not used for any experiments. DAPI or Hoechst 33342 is routinely used to stain 4% PFA fixed cells to objectively screen for mycoplasma by fluorescence microscopy, using 100x objective. We also routinely submit supernatant from our cultured cells to the University of Manchester Faculty of Biology Medicine and Health’s Media Order facility which provides a PCR-based mycoplasma detection service.
12. What was the estimated purity (%) of the primary pancreatic ductal and stellate cell cultures used in this study, and how was this measured?
Authors Response: The current study unfortunately did not estimate the purity of the primary pancreatic ductal and stellate cells used. The hPSC and HPDE were obtained as a kind gift from Prof David I Yule and Prof Diane Simeone, respectively. hPSC were isolated from a PDAC patient undergoing a Whipple procedure and were isolated by a modified method (PMID 21148289) first established by Apte, MV et al (1998; PMID 9771417). The stellate characteristics of hPSC (gifted to us) were previously verified by Won, JH et al (2011; PMID 21148289). On the other hand, HPDE is HPV E6/E7-immortalized (no p53 or RB) and are of ductal origin. The characteristics of HPDE (that were gifted to us) were described by Abel, EV et al (2018; PMID 30074477). hPSC and HPDE cells were used as “non-malignant” pancreatic cells to screen for PMCA isoform expression. However, these cells were not taken forward for any future functional experiments and therefore we felt it unnecessary to characterize their purity. However, we have added further references to include the original sources which these cultures were acquired from in section 2.2 (first paragraph, page 4-5).
13. P. 15/20, lanes 501-513: How was equal loading of samples controlled? Were protein concentration measurements of lysates and, for example Ponceau red staining and densitometry of lanes on blots done, or does control of equal loading rely exclusively on b-actin?
Authors Response: The current study attempted to control for equal protein loading into wells of gels by two key processes: 1) Pre-loading (using Bradford assay to calculate the protein concentration in the sample lysate) and 2) Post-loading (using β-actin as the loading control). All loading samples prepared in a concentration and volume adjusted manner to ensure that every well was loaded with the same sample volume (e.g. 5 ug protein, 20 ul lysate loading volume). Ponceau staining of the nitrocellulose membranes was also routinely performed to ensure efficient transfer from the gel to the nitrocellulose membrane and additional check for protein loading. In addition, the house keeping protein such as β-actin was routinely used to check for any potential loading error. For this study, β-actin was also used for normalisation of protein bands when band density quantification was required.
14. P. 7/20, lane 206: To obtain calcium influx following cyclopiazonic acid treatment, 20 mM Ca2+is added to obtain store operated calcium entry into the cells. This seems to be a rather supra-physiological concentration. What motivated the use of 20 mM Ca2+?
Authors Response: The use of 20 mM Ca2+ during the in situ Ca2+ clearance assay is designed to maximise the Ca2+ entry during activation of store-operated Ca2+ entry (SOCE) following treatment with the SERCA inhibitor, CPA and to increase the dynamic range of [Ca2+]i concentrations over which Ca2+ clearance can be accurately assessed upon removal of external Ca2+ (with EGTA).
15. Authors discuss cyclopiazonic acid-induced effects in the context of cytoplasmic calcium overload induced by store operated calcium entry. However, it should be kept in mind, that cyclopiazonic acid treatment can induce cell stress also due to endoplasmic reticulum calcium depletion (“endoplasmic reticulum stress”).
Authors Response: We agree with the referee and are fully aware that using cyclopiazonic acid (CPA) would induce ER stress. We have addressed this concern in section 2.6 (paragraph 2 and 3; p.10). For the short duration (12 h) of our apoptotic study, we showed that siNT control cells were unaffected by the CPA treatment over the duration of 12 hours whereas the majority of siPMCA4 treated cells underwent caspase3/7-induced apoptosis. Since both sets of cells should undergo the same degree of ER depletion and thus ER stress, but the siPMCA4-treated cells exhibited impaired Ca2+ clearance, Ca2+ homeostasis and elevated resting [Ca2+]i suggests that the increase in apoptosis was cytosolic Ca2+ overload –dependent and not ER stress dependent.
16. Knock down of PMCA4 in this work was partial, reaching an approximately 70% down-regulation of expression, and the resting cytosolic calcium levels of PMCA4 knock-down cells was strongly elevated when compared to control cells (P.9/20, lane 281). Because the calcium transport activity of PMCA enzymes is strongly enhanced by calcium/calmodulin and other mechanisms, it should be noted in the manuscript, that the remaining (30%) PMCA4 enzyme, as well as PMCA1, also present in the cells, were very probably in a strongly stimulated state, and that their stimulated transport activity contributed significantly to the re-establishment of the calcium homeostasis (and hence survival) of the cells after PMCA4 knockdown.
Authors Response: We thank the referee for pointing this out. We have edited our manuscript to include this informative explanation in the discussion section (paragraph 5, P14).
17. Please give more detailed information regarding the PMCA2 and 3-specific antibodies used in this work.
Authors Response: We have added catalogue numbers for antibodies used in this study in the Methods section 4.6. PMCA2 and PMCA3 primary antibodies are polyclonal antibodies derived from rabbits. Synthetic immunogenic peptides were used to generate these primary antibodies.
18. Authors state that “targeting PMCA4 may potentially be beneficial as a therapeutic adjuvant which selectively sensitizes PDAC cells to currently available clinical therapy while sparing healthy tissues” (p. 13/20, lanes 448-449). How could this be accomplished in a clinically meaningful manner?
Authors Response: The current study highlights that PMCA4 is almost uniquely over-expressed in PDAC, which may contribute to oncogenicity and therefore may represent a novel therapeutic target. However, despite attempts, there are currently no PMCA4-selelctive inhibitors currently approved for therapeutic purposes, or undergoing trials. There are some PMCA4 selective inhibitors currently being investigated for a wide range of purposes including Cardiovascular diseases. This study, therefore, may provide an additional rationale for the development of clinical PMCA4 selective inhibitors. However, until these inhibitors become available, the most meaningful way to examine whether targeting PMCA4 may be useful as a therapeutic adjuvant is to combine silencing/knockout of PMCA4 function (using both in vitro 3D organoid models, in vivo xenograft or GM mouse models) currently available PDAC first-line chemotherapeutics (e.g. Gemcitabine, Nab-paclitaxel, cis-platin). This would tell us whether loss of PMCA4 functionally sensitizes PDAC cells to standard chemotherapeutic. Additional combination therapies with more recently developed drugs that target the acidification of the tumour microenvironment (e.g. PKM2, LDH, MCT inhibitors, etc.) may also be beneficial. These have now been included in the Discussion (page 14-15).
19. S2, Panel A (Legend): “Representative TMRE traces are shown with black arrows…” All arrows in this Panel are black (delete "black").
Authors Response: The word black has been removed from S2 legend.
20. Fig. S5 and Legend are somewhat confusing, and colors in this Figure are hard to distinguish. Calculation of R2/R1 should be explained more clearly.
Authors Response: We have removed the Method S2 section for a “paired” Ca2+ clearance comparison to avoid confusion as the current study only employs an “unpaired” Ca2+ clearance experimental design. A bolder line and extra panels showing the exponential fit has been added to clarify the figure. The legend text has been edited to accommodate the changes in Fig S5.
21. Ref. 34 (lanes 692-693) deals with programmed cell death induced by endoplasmic reticulum stress in soybean cells. Is this an adequate reference as used in lane 389 of the Manuscript, for Jurkat cells?
Authors Response: We have now revised the reference as the referee advised. We agree that this reference may not be appropriate in this context and have removed it from the manuscript.
22. Please note that the correct spelling of the PDAC line used in this study as per ATCC or the original paper of Yunis et al., Int. J. Cancer, 1977, is: MIA PaCa-2.
Authors Response: The authors thank the referee for pointing out this error. We have edited the revised manuscript accordingly.
Round 2
Reviewer 1 Report
The authors have clarified several of the questions I raised in my previous review.
I still think that the finding "PMCA4 knockdown sensitizes MIAPaCa-2 cells to apoptosis" and in discussion section open a frame of thinking that is already elicitated within the quoted paper by the laboratory of Prendergast; I think it is important to mention that that particular study refers to APOPTOSIS and the related mechanisms within this particular PDAC model: nonetheless, a tight correlation exists between impaired calcium-glucose metabolism, immune-infiltrate, angiogenesis and cancer progression and dissemination to distant sites and to nodal compartment. Indeed, immune cells come and go across the permeable capillaries. Because of these intimate interactions, the capacity of dendritic cells and endothelial cells ECs as antigen-presenting cells (APC) can be also discussed, since several examples have been recently published (i.e. PMID: 31277479), especially while explaining MIAPaCa-2 cells neoplastic phenotype.
Nonetheless, I acknowledge the authors efforts made to improve the quality of the manuscript, and all revisions can be subjective.
The manuscript reached the proper novelty and scientific soundness required.
Author Response
The authors have clarified several of the questions I raised in my previous review.
I still think that the finding "PMCA4 knockdown sensitizes MIAPaCa-2 cells to apoptosis" and in discussion section open a frame of thinking that is already elicitated within the quoted paper by the laboratory of Prendergast; I think it is important to mention that that particular study refers to APOPTOSIS and the related mechanisms within this particular PDAC model: nonetheless, a tight correlation exists between impaired calcium-glucose metabolism, immune-infiltrate, angiogenesis and cancer progression and dissemination to distant sites and to nodal compartment. Indeed, immune cells come and go across the permeable capillaries. Because of these intimate interactions, the capacity of dendritic cells and endothelial cells ECs as antigen-presenting cells (APC) can be also discussed, since several examples have been recently published (i.e. PMID: 31277479), especially while explaining MIAPaCa-2 cells neoplastic phenotype.
Nonetheless, I acknowledge the authors efforts made to improve the quality of the manuscript, and all revisions can be subjective.
The manuscript reached the proper novelty and scientific soundness required.
Authors Response
As we alluded to in our rebuttal to the original comment by Referee 1, we have never referred to, quoted nor cited any paper from the laboratory of Prendergrast in either the original or revised manuscript. In the context of our specific observation that “PMCA4 knockdown sensitizes MIA PaCa-2 cells to apoptosis”, all cited papers are from the laboratory of Greg Monteith and include the following:
Curry MC, Luk NA, Kenny PA, Roberts-Thomson SJ, Monteith GR. Distinct regulation of cytoplasmic calcium signals and cell death pathways by different plasma membrane calcium ATPase isoforms in MDA-MB-231 breast cancer cells. J Biol Chem. 2012 Aug 17 Aung CS, Kruger WA, Poronnik P, Roberts-Thomson SJ, Monteith GR. Plasma membrane Ca2+-ATPase expression during colon cancer cell line differentiation. 2007 Biochem Biophys Res Commun. 2007 Apr 20;355(4):932-6 Stewart TA, Yapa KTDS, Monteith GR. Altered calcium signaling in cancer cells. Biochim Biophys Acta. 2015;1848(10):2502–11.These papers relate to similar observations in breast/colon cancer cells and are all directly pertinent to our specific observation in PDAC cells. Therefore, we are really struggling to understand what Referee 1 is referring to here in relation to papers authored or co-authored by Prendergast. In a concerted attempt to address this we performed a cursory search of pubmed which revealed that there are 10 papers identified from a search of “Prendergast and pancreatic cancer”. Two of these papers by Prendergast C relate to surgical resection of pancreatic cancer. See below:
Lopez NE, Prendergast C, Lowy AM. Borderline resectable pancreatic cancer: definitions and management. World J Gastroenterol. 2014 Aug 21;20(31):10740-51. PMID: 25152577
Artinyan A, Soriano PA, Prendergast C, Low T, Ellenhorn JD, Kim J. The anatomic location of pancreatic cancer is a prognostic factor for survival. HPB. 2008;10(5):371-6. PMID: 18982154
The above studies correlate tumour location within the pancreas with poor patient survival (body and tail) with distant metastases and reduced harvested lymph nodes, and therefore reduced immune infiltration. These are peripherally relevant to the comments made by referee 1 above but fail to relate to our specific observations in our current study in the context of PMCA4 expression, apoptosis resistance and/or metabolism.
From the original pubmed search of “Prendergast and pancreatic cancer” there were three relevant papers authored by Prendergast GC:
Nevler A, Muller AJ, Sutanto-Ward E, DuHadaway JB, Nagatomo K, Londin E, O'Hayer K, Cozzitorto JA, Lavu H, Yeo TP, Curtis M, Villatoro T, Leiby BE, Mandik-Nayak L, Winter JM, Yeo CJ, Prendergast GC, Brody JR. Host IDO2 Gene Status Influences Tumor Progression and Radiotherapy Response in KRAS-Driven Sporadic Pancreatic Cancers. Clin Cancer Res. 2019 Jan 15;25(2):724-734. PMID: 30266763
Nevler A, Muller AJ, Cozzitorto JA, Goetz A, Winter JM, Yeo TP, Lavu H, Yeo CJ, Prendergast GC, Brody JR. A Sub-Type of Familial Pancreatic Cancer: Evidence and Implications of Loss-of-Function Polymorphisms in Indoleamine-2,3-Dioxygenase-2. J Am Coll Surg. 2018 Apr;226(4):596-603. PMID: 29426021
Witkiewicz AK, Costantino CL, Metz R, Muller AJ, Prendergast GC, Yeo CJ, Brody JR. Genotyping and expression analysis of IDO2 in human pancreatic cancer: a novel, active target. J Am Coll Surg. 2009 May;208(5):781-7; discussion 787-9. PMID: 19476837
These papers relate specifically to the role of the Indoleamine 2,3-dioxygenase (IDO-1) in pancreatic cancer, which is the rate-limiting enzyme that metabolizes tryptophan in the kynurenine pathway, an important mechanism for immune tolerance of tumour cells. These papers may well be what Referee 1 is referring to, although it is not clear. Indeed, we accept and whole heartedly agree that IDO-1 and the general concept of immune tolerance are very important in pancreatic cancer. However, it is very difficult to “shoe-horn in” these concepts into our manuscript in the context of PMCA4 expression/knockdown, apoptosis resistance/sensitization and glycolytic metabolism, which is the mayor focus of our study. Nevertheless, regardless of any paper by Prendergast cited or otherwise, the general comments made by Referee 1 linking “impaired calcium-glucose metabolism, immune-infiltrate, angiogenesis and cancer progression and dissemination to distant sites and to nodal compartment” are insightful and likely highly relevant to pancreatic cancer. We attempted to address this in our response the comment 6 (page 13 of original manuscript) from Referee 1. Specifically, in the first revision we made an important link between PDAC cell metabolism, tumour acidity (lactic acid efflux), PMCA function (ATP-driven Ca2+/H+-exchanger), and immune tolerance. This is because tumor acidity acts as a broad immune escape mechanism by which cancer cells, inhibit anti-tumour immune effectors (including T cells, NK cells and crucial antigen-presenting dendritic cells), while simultaneously promoting the immunosuppressive and pro-tumour properties of regulatory T cells and myeloid cells. Moreover, this acidic microenvironment may also specifically promote PMCA4 activity and thus further promote cell survival. We felt that this was a creative way of linking these seemingly disparate topics of cancer metabolism and PMCA function with immune surveillance and migration/invasion and the consequent metastases that was highly relevant to our study. Therefore, on balance we feel that it would be inappropriate to “shoe-horn in” any additional comments on this topic, and especially related to studies by Prendergast, without significantly losing relevance or focus. We hope that Referee 1 will be sympathetic to this decision.
Reviewer 3 Report
The authors for sure have improved the manuscript. One point remains still open and it’s the lack of a clear translational potential. I feel that this paper is more suitable for a basic-research journal only, but the final judgment is for the editors. I acknowledge at the same time that the efforts made by the authors indicate that they belong to a strong group of research in this field.
Author Response
The authors for sure have improved the manuscript. One point remains still open and it’s the lack of a clear translational potential. I feel that this paper is more suitable for a basic-research journal only, but the final judgment is for the editors. I acknowledge at the same time that the efforts made by the authors indicate that they belong to a strong group of research in this field.
Authors Response
We are pleased that referee 3 found additional merit and improvement in our revised manuscript but regret that he/she still feels that a clear lack of translational potential remains. We respectfully disagree and would like to reiterate our original point that this manuscript is timely and will act as a springboard for future studies not only by our own group but other researchers in the field such that the true translational potential can be realized sooner. We hope that the Editors can make a positive decision.
Reviewer 5 Report
Comments to the revised version:
The issue raised originally in 9. has still not been addressed in the Abstract or in Conclusions. There are PMCA4”high” and PMCA4”low” PDAC cell lines, as well as tumors (which, as Authors note, stratify according to survival). All conclusions regarding function pertain only to "PMCA4high" cells in this study. However, both PMCA4high and low tumors still belong to the category of pancreatic ductal adenocarcinoma. It is not acceptable to equate MIA PaCa-2 (and its high PMCA4 expression) with PDAC en bloc. Stating simply that this issue is “hypothesis generating” (lane 104) is not a sufficiently in-depth way to discuss this issue, which is, after all, rather central to this Manuscript. It would be great, if Authors could attempt to generate such hypotheses. It is possible that in PMCA4low cells (such as PANC-1 or the tumors that express less PMCA4), PMCA1 function is crucial.
Lanes 47-48: the issue of PMCA genes/primary transcripts (4) versus splice isoforms (>20) has still not been corrected.
Lane 136: “clkassical” : classical
Please homogenize: PANC-1 or Panc-1 (lane 156)
Lane 586: The variation of the calcium concentration in the perfusion experiments is not clear: “Dye-loaded cells were mounted onto a perfusion chamber attached with gravity-fed perfusion system (Harvard apparatus, Massachusetts, USA) then perfused with Ca2+-free HPSS containing 1 mM EGTA and 30 μM CPA for 20 minutes. Ca2+ clearance is observed by switching from 20 mM Ca2+ HPSS to Ca2+-free HPSS as previously described.” When is calcium added? Moreover, the use of 20 mM calcium should be explained to readers here.
Fura or fura? (lanes 298, 300, 319)
Lane 656: MIA PaCa-2

Author Response
The issue raised originally in 9. has still not been addressed in the Abstract of in Conclusions. There are PMCA4”high” and PMCA4”low” PDAC cell lines, as well as tumors (which, as Authors note, stratify according to survival). However, these all still belong to the pancreatic ductal adenocarcinoma category. It is not acceptable to equate MIA PaCa-2 (and its high PMCA4 expression) with PDAC en bloc. Stating simply that this issue is “hypothesis generating” (lane 104) is not a sufficiently in-depth way to discuss this issue, which is, after all, rather central in this Manuscript. It would be great, if Authors could attempt to generate such hypotheses.Authors Response
We have now attempted to address these concerns further by adding the following sentence to the Abstract:
“Western blot and RT-qPCR revealed that MIA PaCa-2 cells almost exclusively express PMCA4 making these a suitable cellular model of PDAC with poor patient survival”
In addition we have attempted to incorporate some of the comments by Referee 5 by expanding paragraph 3 in the Discussion with the following text (page 13, lines 416-424; highlighted in green in the second revised manuscript):
“However, it must be noted that PMCA4 is not overexpressed in all PDAC cells (e.g. PANC-1) and PMCA4 abundance may depend on cell differentiation status and confluency [24]. Moreover, PDAC cells are well or moderately differentiated histologically, which may influence PMCA4 abundance. Nevertheless, Kaplan Meier survival analysis suggests that PMCA4 overexpression correlates to poor patient survival. Therefore, MIA PaCa-2 cells constitute a well suited model for an eventual “PMCA4 high expression” molecular subtype of PDAC. Furthermore, by extrapolation MIA PaCa-2 cells may also represent a good cellular model for testing/screening novel putative PMCA4-specific inhibitors in future studies that would be predicted to improve PDAC patient survival.”
Lanes 47-48: the issue of PMCA genes/primary transcripts (4) versus splice isoforms (>20) has still not been corrected.Authors Response
We apologize and have now corrected this in the manuscript (page 2, lines 49-50 highlighted in green) and cited an additional paper (Strehler EE and Zacharias DA (2001); ref 11, highlighted in green).
Lane 136: “clkassical” : classicalAuthors Response
This has now been corrected (page 4, line 137; highlighted in green).
Please homogenize: PANC-1 or Panc-1 (lane 156)Authors Response
This has now been corrected (page 5, line 157; highlighted in green).
Lane 586:The variation of the calcium concentration in the perfusion experiments is not clear: “Dye-loaded cells were mounted onto a perfusion chamber attached with gravity-fed perfusion system (Harvard apparatus, Massachusetts, USA) then perfused with Ca2+-free 586 HPSS containing 1 mM EGTA and 30 μM CPA for 20 minutes. Ca2+ clearance is observed by switching from 20 mM Ca2+ HPSS to Ca2+-free HPSS as previously described.” When is calcium added? Moreover, the use of 20 mM calcium should be explained here.Fura or fura? (lanes 298, 300, 319)
Authors Response
We apologise and thank the referee for identifying this mistake. We have now expanded the description to further clarify this experimental approach with the following text (page 17, lines 594-601; highlighted green):
.”This led to ER Ca2+ store depletion and activation of store operated Ca2+ entry (SOCE) channels. Therefore, subsequent perfusion of cells with HPSS containing high external Ca2+ (20 mM) led to a marked Ca2+ entry and increase in cytosolic Ca2+ ([Ca2+]i) which reached a short-lived steady state due to a balance between Ca2+ entry and Ca2+ efflux. Therefore, subsequent removal of external Ca2+, by perfusion with Ca2+-free/EGTA-containing HPSS, allowed [Ca2+]i clearance to be observed and assessed. The addition of such high external Ca2+ was necessary such that [Ca2+]i clearance rate could be assessed over a much greater dynamic range of [Ca2+]i. These methods have been fully characterized in our previous studies [9,10]”.
We have also corrected Fura-2 to fura-2 in line 586.
Lane 656: MIA PaCa-2Authors Response
This has now been corrected (page 4, line 137; highlighted in green).